# SOCIAL NETWORK STRUCTURE SHAPES INNOVATION: EXPERIENCE SHARING IN RL WITH SAPIENS

## ABSTRACT

Human culture relies on innovation: our ability to continuously explore how existing elements can be combined to create new ones. Innovation is not solitary, it relies on collective search and accumulation. Reinforcement learning (RL) approaches commonly assume that fully-connected groups are best suited for innovation. However, human laboratory and field studies have shown that hierarchical innovation is more robustly achieved by dynamic social network structures. In dynamic settings, humans oscillate between innovating individually or in small clusters, and then sharing outcomes with others. To our knowledge, the role of social network structure on innovation has not been systematically studied in RL. Here, we use a multi-level problem setting (WordCraft), with three different innovation tasks to test the hypothesis that the social network structure affects the performance of distributed RL algorithms. We systematically design networks of DQNs sharing experiences from their replay buffers in varying structures (fully-connected, small world, dynamic, ring) and introduce a set of behavioral and mnemonic metrics that extend the classical reward-focused evaluation framework of RL. Comparing the level of innovation achieved by different social network structures across different tasks shows that, first, consistent with human findings, experience sharing within a dynamic structure achieves the highest level of innovation in tasks with a deceptive nature and large search spaces. Second, experience sharing is not as helpful when there is a single clear path to innovation. Third, the metrics we propose, can help understand the success of different social network structures on different tasks, with the diversity of experiences on an individual and group level lending crucial insights.

## 1 INTRODUCTION

Unlike herds or swarms, human social networks solve different tasks with different topologies (Momennejad, 2022). Human and computational studies show that properties of both the social network structure and task affect the abilities of groups to search collectively: social network structures with high connectivity are better suited for quick convergence in problems with clear global optima (Coman et al., 2016; Momennejad et al., 2019), while partially-connected structures perform best in *deceptive* tasks, where acting greedily in the short-term leads to missing the optimal solution (Derex & Boyd, 2016; Lazer & Friedman, 2007; Cantor et al., 2021; Du et al., 2021; Adjodah et al., 2019). Despite this evidence, works in distributed reinforcement learning (RL) have focused on fully-connected architectures (Mnih et al., 2016; Horgan et al., 2018; Espeholt et al., 2018; Nair et al., 2015; Christianos et al., 2020; Schmitt et al., 2019; Jaderberg et al., 2018). Here, we test the performance of different social network structures in groups of RL agents that share their experiences in a distributed RL learning paradigm. We refer to such groups as multi-agent topologies, introduce SAPIENS, a learning framework for Structuring multi-Agent toPologies for Innovation through ExperieNce Sharing[1], and evaluate it on a deceptive task that models collective innovation.

Innovations represent the expansion of an agent's behavioral repertoire with new problem-solving abilities and are, therefore, a necessary ingredient of continuous learning (Leibo et al., 2019). They arise from tinkering, recombination and adoption of existing innovations (Solé et al., 2013; Derex & Boyd, 2016) and have been characterized as a type of combinatorial search constrained by semantics

---

[1]We provide an implementation of SAPIENS and code to reproduce the simulations we report.

dictating the feasible combinations of innovations (Solé et al., 2013; Derex & Boyd, 2016). We adopt this definition: innovations are a type of collective search task with : a) a multi-level search space, where innovations arise out of recombination of existing ones (Hafner, 2021) b) rewards that increase monotonically with the level of innovation, in order to capture the human intrinsic motivation for progress (Solé et al., 2013).

Laboratory and field studies of human groups have shown that collective innovation is highly contingent on the social network structure (Momennejad, 2022; Migliano & Vinicius, 2022; Derex & Boyd, 2016). The reason for this lies in the exploration versus exploitation dynamics of social networks. High clustering and long shortest path in partially-connected structures help maintain diversity in the collective at the benefit of exploration, while high connectivity quickly leads to conformity, which benefits exploitation (Lazer & Friedman, 2007). Of particular interest are structures that achieve a balance in this trade-off: small-worlds are static graphs that, due a modular structure with long-range connections, achieve both high clustering and small shortest path (Watts & Strogatz, 1998). Another example are *dynamic structures*, where agents are able to periodically change neighbors (Volz & Meyers, 2007). These two families of graphs have the attractive property that they both locally protect innovations and quickly disseminate good solutions (Derex & Boyd, 2016).

Despite progress on multiple fronts, many open questions remain before we get a clear understanding of how social network structure shapes innovation. On the cognitive science side, computational and human laboratory studies of collective innovation are few and have studied a single task where two innovations are combined to create a new one (Derex & Boyd, 2016; Cantor et al., 2021), while most works study other types of collective search that do not resemble innovation (Mason & Watts, 2012; Mason et al., 2008; Lazer & Friedman, 2007; Fang et al., 2010). Furthermore, laboratory studies have collected purely behavioral data (Mason et al., 2008; Derex & Boyd, 2016), while studies of collective memory have shown significant influence of social interactions on individual memory (Coman et al., 2016), indicating that mnemonic data may be another good source of information. In distributed RL, studies are hypothesizing that the reason why groups outperform single agents not just in terms of speed, but also in terms of performance, is the increased diversity of experiences collected by heterogeneous agents but not explicitly measure it (Nair et al., 2015; Horgan et al., 2018). In this case two steps seem natural: introducing appropriate metrics of diversity, and increasing it, not only through heterogeneity, but also through the social network topology.

To achieve this we propose SAPIENS, a distributed RL learning framework for modeling a group of agents exchanging experiences according to a social network topology. We study instantiations of SAPIENS where multiple DQN learners (Mnih et al., 2013) share experience tuples from their replay buffers with their neighbors in different static and dynamic social network structures and compare them to other distributed RL algorithms (Mnih et al., 2016; Nair et al., 2015). We employ Wordcraft (Jiang et al., 2020b) as a test-bed and design three custom tasks (Figures 1 and 2) covering innovation challenges of different complexity: (i) a task with a single innovation path to an easy-to-find global optimum. This type of task can be used to model a linear innovation structure, such as the evolution of the fork from knife to having, two-, three- and, eventually four tines. (Solé et al., 2013) and is not a deceptive task. (ii) a task with two paths that individually lead to local optima, but when combined, can merge toward the global optimum. Ispired from previous studies in cognitive science (Derex & Boyd, 2016; Cantor et al., 2021) this task we can capture innovations that were repurposed after their invention, such as Gutenberg's screw press leading to the print press (Solé et al., 2013). (iii) a task with ten paths, only one of which leads to the global optimum, which captures search in vast spaces. In addition to the two deceptive tasks in Wordcraft, we also evaluate SAPIENS algorithms on a deceptive task implemented in a grid world. We empirically show that the performance of SAPIENS depends on the inter-play between social network structure and task demands. Dynamic structures perform most robustly, converging quickly in the easy task, avoiding local optima in the second task, and exploring efficiently in the third task. To interpret these findings, we propose and compute novel behavioral and mnemonic metrics that quantify, among others, the diversity of experiences.

**Contributions** Our contributions are two-fold. **From a cognitive science perspective**, SAPIENS is, to our knowledge, the first computational study of hypotheses in human studies relating social network structure to collective innovation, that employs deep RL as the individual learning mechanism. Compared to the simple agent-based models employed by previous computational studies (Lazer & Friedman, 2007; Cantor et al., 2021; Mason & Watts, 2012), deep RL offers three main advantages : i) it enables empirical experiments with more complex test-beds and larger search spaces

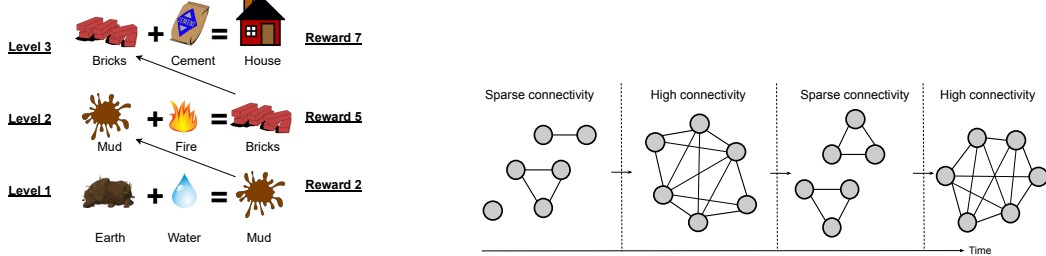

Figure 1: (Left) Illustration of an innovation task, consisting of an initial set of elements (Earth, Water) and a recipe book indicating which combinations create new elements. Upon creating a new element the player moves up an innovation level and receives a reward that increases monotonically with levels. (Right) Dynamic social network structures oscillate between phases of low connectivity, where experience sharing takes place within clusters, and high connectivity, where experiences spread between clusters.

ii) agents can share their experience by simply exchanging transitions from their respective replay buffers, without requiring ad-hoc mechanisms for copying the behaviors of other agents, such as the majority rule (Lazer & Friedman, 2007; Cantor et al., 2021) iii) by using the replay buffer as a proxy of the memories of agents, we can directly measure properties such as the diversity of experiences that are challenging to estimate with humans. Aside these methodological contributions, as we will see later, our empirical study leads to clear hypotheses for future experiments with humans. **From an RL perspective**, our work extends upon the distributed RL paradigm by systematically analyzing the effect of static and dynamic social network structures on different types of innovation tasks both in terms of performance and novel behavioral and mnemonic metrics.

## 2 METHODS

### 2.1 WORDCRAFT: A TEST-BED FOR INNOVATION

We perform experiments on Wordcraft (Jiang et al., 2020b), an RL environment inspired from the game Little Alchemy 2 [2]. As we illustrate on the left of Figure 1, tasks start with an initial set of elements and the player explores combinations of two elements in order to create new ones.

In Wordcraft one can create different types of tasks using a *recipe book* ($\mathcal{X}_{\text{valid}}$), which is a dictionary containing the valid element combinations of two elements and the newly crafted one. In addition to the recipe book, a task requires a reward function $R_{\text{valid}}$ that returns a scalar reward associated with crafting element $z$. A valid combination returns a new element and reward only the first time it is chosen. When queried with a non-valid combination, $\mathcal{R}_{valid}$ returns a reward of zero. Thus, a task can be be described by a tuple $(\mathcal{X}_0, \mathcal{X}_{valid}, \mathcal{R}_{valid}, T)$, where $\mathcal{X}_0$ denotes the initial set of elements and $T$ is the number of time steps available to an agent before the environment resets, and can be modelled as a fully-observable Markov Decision process (see Appendix A).

### 2.2 INNOVATION TASKS

We introduce the following concepts to characterize the structure of innovation tasks. An innovation task can contain one or more paths, which can potentially be connected to each other (as in the merging paths task). In our proposed tasks, an **innovation path** $X$ is defined as a sequence of elements $[X_1, ..., X_n]$, where crafting an element $X_i$ $(i > 1)$ requires to combine the previously crafted element $X_{i-1}$ and a base element from the initial set. The first element $X_1$ in the sequence requires combining elements from the initial set or from other paths. The **innovation level** of an element corresponds to the length of the path it belongs to, plus the sum of the path lengths required to craft its first item (0 if the first item is crafted from base elements). For example, the innovation level of $A_3$ in the single path task is 3, while the innovation level of element $C_1$ in the merging paths task is 5 (1 on C + 2 on A + 2 on B). Within an innovation task, the **trajectory** of an agent is defined

---

[2]https://littlealchemy2.com/.

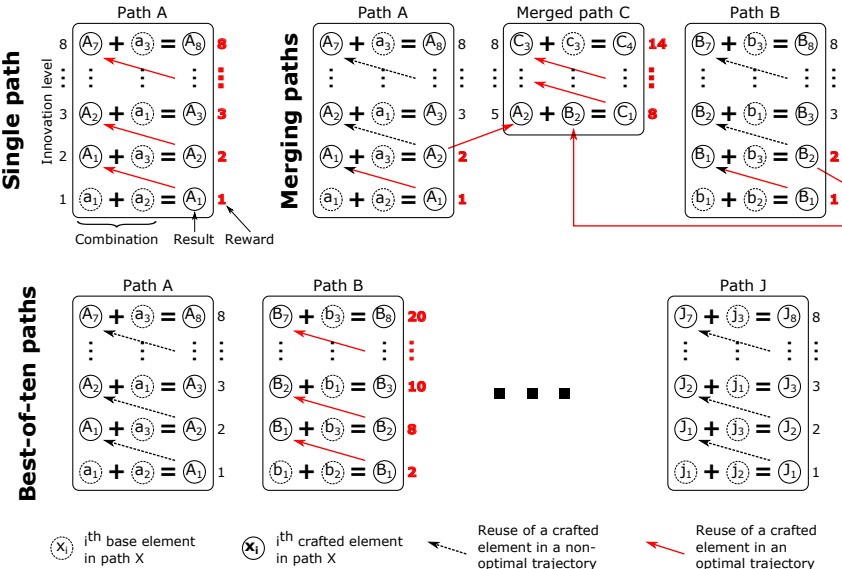

Figure 2: We introduce three innovation tasks called single path, merging paths and best-of-ten paths, described in Section 2.2. Each task contains one or more paths, labeled by an uppercase letter ($A$ to $J$). Each path $X$ has its own initial set of three base elements $\{x_1, x_2, x_3\}$, which are represented in dashed circles. Crafted elements in path $X$ are represented in upper case ($X_i$) in solid circles. Optimal trajectories for each tasks are represented by solid red arrows, with their corresponding reward in bold red.

as the sequence of crafted elements it produces. Finally, the **optimal trajectory** of an innovation task is the trajectory that returns the highest cumulative reward within the problem horizon $T$. We design three innovation tasks, shown in Figure 2, that pose different challenges:

**Single innovation path**    The initial set contains three elements ($\mathcal{X}_{valid} = \{a_1, a_2, a_3\}$) and . An agent needs to first combine two of them to create the first element and then progresses further by combining the most recently created element with an appropriate one from the initial set. This optimization problem contains a single global optimum.

**Merging paths**    There are two paths, A and B, and at level 2 there is a cross-road that presents the player with three options: moving forward to the end path A, moving forward to the end of path B, or combining elements from path A and B to progress on path C. The latter is more rewarding and is, thus, the optimal choice. This task is particularly challenging because the player needs to avoid two local optima before finding the global optimum.

**Best-of-ten paths**    Here, one of the ten paths is the most rewarding, but, to find it, the player must first explore and reject the other nine paths. This optimization task is characterized by a single global optimum and 9 local one and its challenge lies in its large search space.

## 2.3 LEARNING FRAMEWORK

SAPIENS considers a group of $K$ DQN agents, where each agent interacts with its own copy of the environment and can share experiences with others. An undirected graph $\mathcal{G}$, with nodes indicating agents and edges indicating that two agents share experiences with each other, determines who shares information with whom . We define the neighborhood of agent $k$, $\mathcal{N}_k$, as the set of nodes connected to it. At the end of each episode the agent shares experiences with each of its neighbors with probability $p_s$: sharing consists of sampling a subset of experiences of length $L_s$ from its own buffer $B_k$ and inserting it in the buffers of all neighbors $B_n, n \in \mathcal{N}_k$. Thus, an agent communicates distinct experiences with each of its neighbors. We present a schematic of two DQN agents sharing

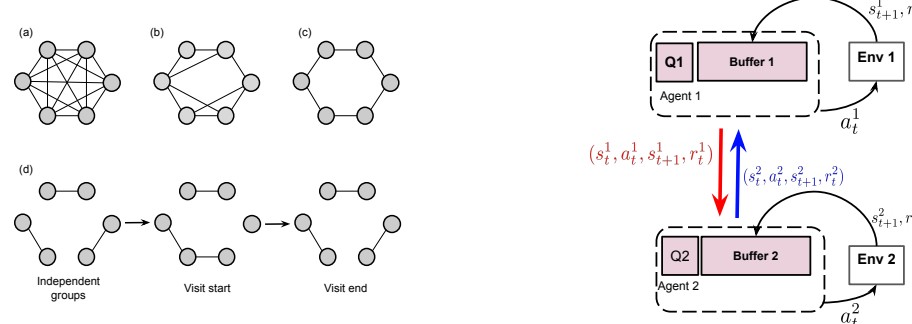

Figure 3: (Left) Social network structures (a) fully-connected (b) small-world (c) ring (d) dynamic. (Right) Schematic of two neighboring DQNs sharing experiences: agent 1 shares experiences from its own replay buffer to that of agent 2 (red arrow) and vice versa (blue arrow) while both agents are independently collecting experiences by interacting with their own copy of the environment.

experiences on the right of Figure 3 and the pseudocode of SAPIENS in Appendix C. We also provide more implementation details about the DQN in Appendix A.

Thus, SAPIENS is a distributed RL learning paradigm where all agents are both actors and learners, a setting distinct from multi-agent RL (Garnelo et al., 2021; Christianos et al., 2020; Jiang et al., 2020a), where agents co-exist in the same environment and from parallelised RL (Steinkraus et al., 2005), where there need to be multiple agents. It should also be distinguished from distributed RL paradigms with a single learner and multiple actors (Horgan et al., 2018; Espeholt et al., 2018; Nair et al., 2015; Garnelo et al., 2021), as multiple policies are learned simultaneously.

We visualize the social network structures studied in our work on the left of Figure 3: fully-connected, small-world, ring and dynamic. We construct the latter by grouping the agents in pairs and then allowing agents to visit other groups for a pre-determined duration, $T_v$ with probability $p_v$. This is a common type of dynamic topology that has been used in human laboratory and field studies (Derex & Boyd, 2016; Migliano & Vinicius, 2022) and thoeretically studied (Volz & Meyers, 2007). We provide more information about it in Appendix D, where we study how its behavior changes with different values of its hyper-parameters $T_v, p_v$. We also present and analyze an alternative type of dynamic topology where the group oscillates between phases of full and no connectivity.

## 2.4 EVALUATION FRAMEWORK

During evaluation trials with experience sharing deactivated, we measure the quality of the final solution and the convergence speed. In addition, we define metrics that are not directly related to performance but can help us analyze the effect of social network structure. These are *behavioral metrics*, characterizing the policies followed by the agents, and *mnemonic metrics*, characterizing the replay buffers of agents, either at an individual or group level.

**Performance-based metrics:** (i) $\mathcal{S}$, group success, a binary variable denoting whether at least one agent in a group found the optimal solution (ii) $R_t^+$: the maximum reward of the group at training step $t$; (iii) $R_t^*$: the average reward of the group at training step $t$; (iv) $T^+$, Time to first success: the first training step at which at least one of the agents found the optimal solution; (v) $T^*$, Time to all successes: the first training step at which all of the agents found the optimal solution (vi) $T^>$, Spread time: number of training steps required for the optimal solution to spread to the whole group, once at least one member discovered it (equals $T^* - T^+$).

**Behavioral metrics:** (i) conformity $C_t$ denotes the percentage of agents in a group that end up with the same element at the end of a given evaluation trial. Thus, agents conform to each other even if they follow alternative trajectories.; (ii) volatility $V_t$ is an agent-level metric that denotes the cumulative number of changes in the trajectory followed by an agent across episodes.

**Mnemonic metrics:** (i) diversity $D_t^k$ is an agent-level metric that denotes the number of unique experiences in an agent's replay buffer; (ii) $D_t^{\mathcal{G}}$ is a group-level metric that captures the diversity of the aggregated group buffer..

## 3 RESULTS

We now evaluate the social network structures presented in Figure 3 on the innovation tasks described in Section 2.2 and visualized in Figure 2. Specifically, methods ring, small-world, dynamic and fully-connected are instantiations of SAPIENS for different social network structures with 10 DQNs where shared experiences are sampled randomly from the replay buffers. We benchmark SAPIENS against: a) no-sharing, a setting with 10 DQN agents without experience sharing b) single, a single DQN agent c) A2C, a distributed policy-gradient algorithm where 10 workers share gradients with a single (Mnih et al., 2016) and d) Ape-X, a distributed RL algorithm with 10 workers that share experience samples with a single DQN learner (Horgan et al., 2018). To test for statistical significance we perform ANOVA for multiple and Tukey range tests for pairwise comparisons. We refer readers to Appendix E for more information about our experimental setup, including tables 2, 2 and 2 that contain the means and standard deviations for evaluation metrics in the three tasks.

### 3.1 OVERALL COMPARISON

In Figure 4 we compare methods across tasks in terms of group success $S$ and Time to first success $T^+$, where we also indicate statistically significant pairwise comparisons with asterisks (more asterisks denote higher significance).

We observe that performance varies across tasks and topologies. The single path task is optimally solved by all methods, except for single and Ape-X, which failed in $20\%$ and $15\%$ of the trials respectively due to slow convergence. ANOVA showed no significant difference in terms of group success $S$ ($p = 0.43$) but methods differed significantly in terms of $T^+$ ($p = 0.2e^{-13}$). In particular, single is significantly slower than all other methods ($T^+ = 648e^3$ for single) and A2C is significantly quicker than other methods ($T^+ = 36200$ for A2C), with the Tukey's range test indicating significance with $p = 0.001$ for all pairwise comparisons with these two methods. This is in agreement with the expectation that a single agent learns slower due to having less data and that the policy-gradient algorithm A2C is more sample efficient than value function algorithms like DQNs. In the merging paths task there were significant differences among methods both for group success $S$ ($p = 0.4e^{-4}$) and convergence speed $T^+$ ($p = 0.0095$). The group success of dynamic ($S = 0.65$) is significantly higher compared to Ape-X ($S = 0.05, p = 0.001$), A2C ($S = 0.0, p = 0.00101$), fully-connected ($S = 0.0, p = 0.00101$) and ring ($S = 0.2, p = 0.0105$). The single, no-sharing and small-world structures performed comparably well, but did not show statistically significant differences with other methods. In terms of $T^+$, we see that dynamic is quicker than other methods with positive $S$, which leads to it being the only method with statistically significant differences with Ape-X ($p = 0.0292$), fully-connected ($p = 0.0421$) and A2C ($p = 0.0421$), the methods that failed in almost all trials and hence have $T^+$ equal to the budget of the experiment. Thus, our main conclusion in the merging-paths task is that methods with fully-connected topologies (fully-connected, A2C, Ape-X) fail dramatically and that the dynamic structure succeeds with higher probability. Finally, in the best-of-ten paths task, differences are also significant both for $T^+$ ($p = 0.9e^{-7}$) and $S$ ($p = 0.15e^{-6}$). Here, dynamic outperforms all methods in terms of $S$ with all Tukey range tests indicating significance with $p = 0.001$, which leads to it also having significantly higher convergence rate ($T^+ = 14e^6$ for dynamic) compared to the other methods that exhausted their time budget.

In additional experiments we have also: a) observed that these conclusions are consistent across group sizes, with increasing sizes leading to better performance in partially-connected and worse in fully-connected structures (see Appendix E.5) b) observed a drop in performance under prioritized experience sharing (Souza et al., 2019; Horgan et al., 2018), where the DQNs employ prioritized replay buffers (Schaul et al., 2016) and experiences with higher priority are shared more often (see Appendix E.6). In agreement with previous works (Souza et al., 2019), we observe that performance degrades in all methods that share experiences. This does not happen for no-sharing, which indicates that prioritized experiences are detrimental only when they are shared. To address this, agents can recompute priorities upon receiving them from other agents to ensure they agree with their own experience (Horgan et al., 2018). c) analyzed how the performance of dynamic varies with its hyper-parameters and derived suggestions for tuning it (see Appendix D) d) monitored the robustness of social network structures to different learning hyper-parameters and observed that dynamic is more robust than fully-connected and no-sharing (Appendix E.4) e) measured alignment of experiences within and across groups that further support our hypothesis that the content of replay buffers

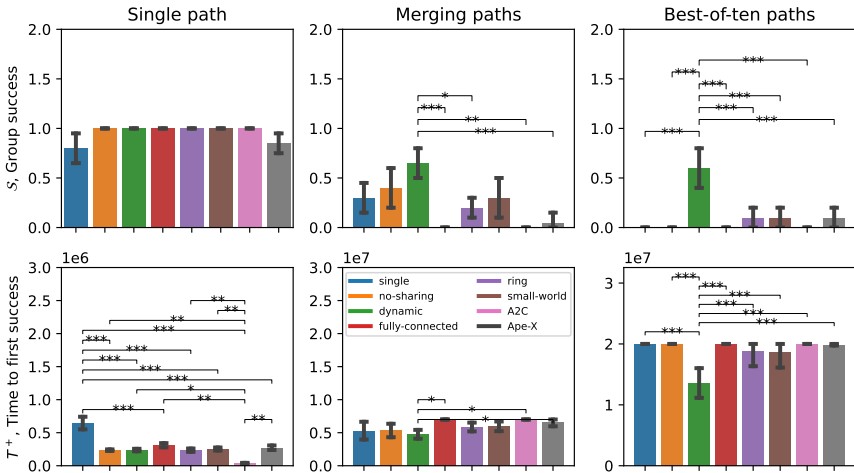

Figure 4: Overall performance comparison for the single path (first column), merging paths (second column) and best-of-ten paths (third column) task in terms of group success ($\mathcal{S}$) (top row) and Time to first success ($T^+$) (bottom row).

is highly contingent on the social network structure (see Appendix E.3) f) performed simulations in another deceptive test-bed and derived similar conclusions to what we observed in Wordcraft, namely that partially-connected structures are better at avoiding local optima (See Appendix E.7) g) tested for the robustness of SAPIENS methods to the amount of sharing (hyper-parameters $L_s$ and $p_s$ introduced in Section 2.3) in Appendix E.8, where we observe sub-optimal performance for low amounts of sharing in dynamic and for large shared batches in fully-connected structures. To explain the differences for SAPIENS under different structures we now study each task in isolation for no-sharing, fully-connected, ring, small-world and dynamic, focusing on the behavioral and mnemonic metrics described in Section 2.4.

## 3.2 TASK: SINGLE PATH

Previous human (Mason et al., 2008) and computational (Lazer & Friedman, 2007) studies have indicated that, when it comes to problems with a single global optimum, fully-connected topologies perform best. Our analysis here, however, indicates no statistically significant difference between methods: the fully-connected topology was actually the slowest ($T^+ = 311e^3$). To shed light into this behavior, we turn towards diversity and volatility. We compare average individual diversity ($\bar{D}_t$) and group diversity $D_t^{\mathcal{G}}$, where for each method we create two samples by looking at these two metrics at the timestep at which diversity starts decreasing. ANOVA tests indicate that differences are significant both for $\bar{D}_{T^+}$ ($p = 0.16e^{-8}$), where all pairwise comparisons are significant based on the Tukey range test, and $D_t^{\mathcal{G}}$ ($p = 0.0005$), where the significantly different pairs are (no-sharing, dynamic, $p = 0.02$), (dynamic, fully-connected, $p = 0.001$), (dynamic, small-world, $p = 0.0087$), (fully-connected, ring, $p = 0.008476$). Intuitively, fully-connected exhibits the highest average individual diversity $\bar{D}_t$ (left in Figure 5) and the lowest group diversity $D_t^{\mathcal{G}}$ (left in Figure 5): sharing experiences with others diversifies an individual's experiences but also homogenizes the group.

This task does not require group diversity, but we expect that high individual diversity should be indicative of quicker exploration. So why does the higher $\bar{D}_t$ of fully-connected not act to its benefit? To answer this we turn to volatility $V_t$ (top left of Figure 5) and observe that it is highest for the fully-connected topology. We tested for statistically significant differences among methods in terms of volatility at convergence ($V_{T_{\text{train}}}$) and found that these pairs differ significantly: (small-world, ring, $p = 0.002387$), (small-world, dynamic, $p = 0.001$), (small-world, no-sharing, $p = 0.001$), (small-world, fully-connected, $p = 0.001$), (ring, fully-connected, $p = 0.001$), (dynamic, fully-connected, $p = 0.001$) and (no-sharing, fully-connected, $p = 0.001$). We hypothesize that the higher individual diversity and volatility of the fully-connected structure are linked and indicate that, although experience sharing helps speed up exploration, it also destabilizes agents, so that no

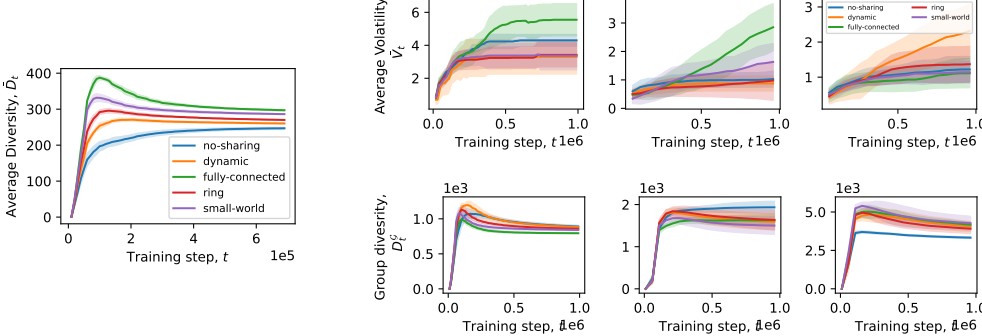

Figure 5: Analyzing group behavior: (Left) Average Diversity $\bar{D}_t$ in the single path task (Right) Average volatility ($V_t$) (top row) and Group Diversity $D_t^{\mathcal{G}}$ (bottom row) in the single path task (first column), merging paths task (second column) and best-of-10 paths task (third column) in term sof

net gain is observed in terms of convergence rate. Notably the fact that experience sharing degrades performance in RL has been observed but not understood (Souza et al., 2019; Schmitt et al., 2019), while as we see here, this becomes possible by analyzing diversity and volatility.

## 3.3 TASK: MERGING PATHS

The hypothesis that motivated this task was that partially connected groups will perform better than fully-connected structures due to their ability to explore diverse trajectories and avoid the two local optima (Derex & Boyd, 2016). This was indeed the case as we saw in our discussion of Figure 4 in Section 3.1. If we also look at the diversities in this task, then we see that the group diversity $D^{\mathcal{G}}$ of fully connected (illustrated in the middle bottom plot in Figure 5) differs significantly to the one of (no-sharing, $p = 0.001$), (ring, $p = 0.0067$) and dynamic ($p = 0.0177$). The group diversity in small-world, which was relatively successful ($S = 0.3$), follows that of the other partially connected structures but peaks at a lower level ($D^{\mathcal{G}} \approx 1700$). We believe that this is due to the dual ability of this topology to both protect and spread information and that its diversity and performance will improve as we increase the size of the group, enabling higher clustering (we illustrate this behavior in Figure 15 in Appendix E.5 where we increase the size of groups to 50 agents).

## 3.4 TASK: BEST-OF-TEN PATH

Which social network topology works best in large search spaces? Our analysis in Figure 4 clearly indicates that dynamic achieves the highest performance. Differently from the two previous tasks where fully-connected exhibited the highest volatility, dynamic and ring, the topologies with the lowest number of connections, are the most volatile here (top row of Figure 5), as agents in them are exploring quickly and, hence, performing better. In terms of group diversity, we found that no-sharing scored significantly lower than all other methods with $p = 0.001$, the small-world surpassed all methods, with its difference from dynamic being marginally statistically significant and fully-connected did not score last. Thus, differently from the other two tasks with small search spaces, quick spreading of information increases group diversity here. However, to solve the task quickly, group diversity needs to be combined with quick, local exploitation of the optimal path, which is possible under structures with large shortest path, such as dynamic and ring. Another interesting observation here and in the single-path task is that dynamic is the only structure that exhibits significantly higher group diversity than no-sharing, indicating that it fosters group exploration.

## 4 RELATED WORK

In distributed RL, shared information has the form of experience tuples or gradients, with the former being preferred due to instabilities and latency concerns in the latter (Horgan et al., 2018; Mnih et al., 2016; Schmitt et al., 2019; Nair et al., 2015; Garnelo et al., 2021). A common social network struc-

ture is that of multiple actors and a single learner (Mnih et al., 2016; Horgan et al., 2018; Schmitt et al., 2019; Garnelo et al., 2021), while the Gorila framework (Nair et al., 2015) is a more general multi-learner architecture, that differs from ours as agents are sharing parameters. Networked Evolutionary Strategies (NetES) consider the effect of network structure on evolutionary strategies (Adjodah et al., 2019), where multiple actors share gradients with a single learner. NetES was applied on continuous control tasks that differ from our deceptive and discrete innovation tasks. In MARL, social network structure determines who co-exists with whom in the environment (Garnelo et al., 2021) or which agents in its neighborhood an agent attends to (Jiang et al., 2020a; Du et al., 2021). Here, dynamic topologies that are adapted to maximize a group's reward have been shown to maximize strategic diversity (Garnelo et al., 2021) and help the agents coordinate on demand (Du et al., 2021). In contrast, our dynamic topologies vary periodically independently of the group's performance, which is important for avoiding local optima. In population-based training, policies are compared against the whole population, thus only considering a fully-connected social network structure (Jaderberg et al., 2018). Admittedly, the literature on the effect of social network structure on collective search is diverse, with different fields making different design choices; to illustrate this we provide a non-exhaustive summary of our literature review in Table 1 of Appendix B.

## 5 Discussion and Future Work

We tested the hypothesis that the social network structure of experience sharing can shape the performance of a group of RL agents using our proposed learning framework, SAPIENS, and showed that, in line with human studies, both social network topology and task structure affect performance. Based on our experimental results, we can provide general recommendations on which topology to use for which task class. In the single-path task, an instance of a class of tasks with no strong local optima (similarly to long-horizon tasks (Gupta et al., 2019)), our results show no benefit of experience sharing. In the merging-path task which exhibits strong local optima that have to be explored up to a certain point in order to discover the global optimum (in the spirit of hard exploration tasks (Baker et al., 2022; Ecoffet et al., 2021)), our results show that topologies with low initial connectivity (such as no-sharing, small world and dynamic) perform best. The dynamic topology shows the highest performance, allowing different groups to explore non-optimal paths before sharing their experience during visits to other groups to find the optimal one. Finally, in the case of large search space with many local optimas (in the spirit of combinatorial optimization tasks (Mazyavkina et al., 2021)), our results show that the dynamic topology performs best, allowing different groups to first explore different paths, then spread the optimal solution to other groups once discovered.

When adopting RL algorithms as computational models for replicating experiments with humans, one needs to acknowledge that their communication and decision-making mechanisms may not faithfully replicate the ones used by humans. One notable difference is that humans may exhibit normative behavior, adopting information not for its utility in the task but for social approval (Mason et al., 2008). From an RL perspective, our study is limited in including experiments only in a few symbolic tasks and a simple navigation task; in the future we plan to study more complex environments like Crafter (Hafner, 2021).

We hope that our work will contribute to the fields of cognitive science and DRL in multiple ways. First, our empirical observations in the single path and best-of-ten-path tasks provide concrete hypotheses for future experiments studying human innovation, which has so far been studied only in a task that inspired our merging-paths task (Derex & Boyd, 2016). By continuing the dialogue that has been initiated between human and computational studies (Fang et al., 2010; Lazer & Friedman, 2007; Cantor et al., 2021) to include DRL methods, we believe that cognitive science will benefit from tools that, as we show here, can learn in realistic problem set-ups and can be analyzed not just in terms of their behavior, but also in terms of their memories. Second, we hope that studies in distributed RL will extend their evaluation methodology by analyzing not just rewards, but also behavioral and mnemonic metrics such as diversity, conformity and volatility that, as we show here, correlate with success. Aside this, the effect of social network structure in distributed RL can be extended beyond evolutionary strategies (Adjodah et al., 2019) and beyond our current instantiation of SAPIENS, by considering other off-policy algorithms than DQNs and other types of information sharing. Finally, considering the effectiveness of the dynamic topologies observed in this study, we envision future works that investigate more types of them, as well as meta-learning or online-adaptation algorithms where the social network structure is optimized for a desired objective.

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

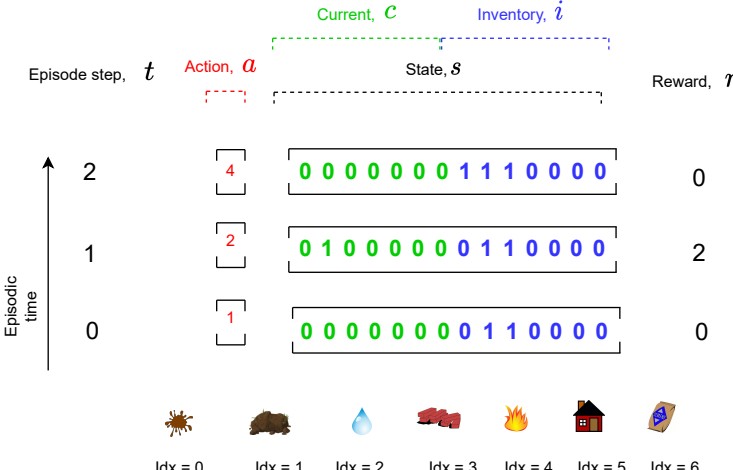

Figure 6: Visualizing actions and states in Wordcraft: we present the first 3 time steps of an episode corresponding to playing the example in Figure 1. This task contains 7 elements, so the action space is a integer with maximum value 7. In the components current $c$ and inventory $i$, each digit in the vector corresponds to the element with the corresponding index. The initial set includes *Water* and *Earth* (their indexes at $\tau = 0$ in the inventory are non-zero). The agent first picks *Earth* (second index in the action vector). At $t = 1$, *Earth* becomes active in the *Current* vector of the state, the the agent selects *Water* and receives a positive reward. At $t = 2$, *Mud* is created and inserted in the inventory and $c$ is cleared.

This supplementary material provides additional methods, results and discussion, as well as implementation details.

- Section A describes in detail the MDP formulation of Wordcraft;
- Section C contains the pseudocde of SAPIENS;
- Section D explains how we model dynamic social network structures and how their performance varies with their hyper-paramaters;
- Section E provides more information about our experimental setup and results (effect of group size, intra-group and inter-group alignment, robustness to learning hyper-parameters and effect of prioritized experience sharing). We also provide tables and figures for all metrics presented in Section 2.4 and reward plots.
- Section E.7 contains simulations with another testbed, the Deceptive Coins game.

## A  DETAILS OF WORDCRAFT AS A MARKOV DECISION PROCESS

We consider the episodic setting, where the environment resets at the end of each episode and an agent is trained for $E_{train}$ episodes. At each time step $t$, the agent observes the state $s_t$ and selects an action $a_t$ from a set of possible actions $\mathcal{A}$ according to its policy $\pi^\theta$, where $\pi^\theta$ is a mapping from states to actions, parameterized by a neural network with weights $\theta$. In return, the agent receives the next state $s_{t+1}$ and a scalar reward $r_t$. Each DQN agent collects experience tuples of the form $[s_t, a_t, s_{t+1}, r_t]$ in its replay buffer.

Figure 6 offers a visualization of the states and actions encountered during an episode in Wordcraft, where the chosen actions and elements are chosen so as to reproduce the example of Figure 1. In order to solve the innovation task described in Section 2.1 we compute the maximum number of elements a player can craft within horizon $T$ for recipe book $\mathcal{X}_{valid}$ and initial set $\mathcal{X}_0$, which we denote as $|X|$. We, then, encode each element as an integer in $[0, |X|)$. Thus, the action space is

| Work | Field | Agent Model | Information type | Task | Dynamic structure? | Main conclusion |
|---|---|---|---|---|---|---|
| (Garnelo et al., 2021) | MARL | DRL | interaction [3] | strategic micro-management (Star-Craft(Vinyals et al., 2017)) | Yes | Topologies with cycles encourage strategic diversity and dynamic ones perform robustly across tasks |
| (Adjodah et al., 2019) | Dec-RL | DRL | rewards, NN weights | continuous control (Mujoco (Todorov et al., 2012)) | No | Random topologies outperforms fully-connected ones |
| (Du et al., 2021) | MARL | DRL | observations | cooperative navigation (Particle World (Lowe et al., 2017)) | Yes | Agents choose to communicate when they need to coordinate. |
| (Dubova et al., 2020) | MARLC [4] | DRL | interaction [1] | coordination game | No | Global connectivity leads to shared and symmetric protocols, while partially-connected groups learn local dialects. |
| (Fang et al., 2010) | computational cognitive science | belief-majority rule [5] | belief, reward | NK problem [6] | No | Partial connectivity maximizes performance |
| (Lazer & Friedman, 2007) | computational cognitive science | belief-majority rule [3] | belief, reward | NK type [7]4 | No | Partial connectivity maximizes performance |
| (Cantor et al., 2021) | computational cognitive science | belief-majority rule [3] | belief, reward | innovation | No | Performance depends on both task and group structure, no topology is robustly optimal across tasks. |
| (Mason & Watts, 2012) | cognitive science | human | action,reward | NK problem [3] | No | Full connectivity maximizes diversity and works best even in complex tasks. |
| (Mason et al., 2008) | cognitive science | human | action, reward | line search | No | Partial connectivity works best in complex problems |
| (Derex & Boyd, 2016) | cognitive science | | action,reward | innovation | Yes | partial connectivity works best |
| (this work) | distributed RL and computational cognitive science | DRL | transition tuples | innovation | yes | Partially-connected structures, especially dynamics ones, perform robustly in different types of innovation tasks |

Table 1: A non-comprehensive summary of the literature on the topic of the effect of social network topology on collective search

$\mathcal{A} = [0, |X|)$, with action $a_t$ indicating the index of the currently chosen element. The state $s_t$ contains two sets of information: a binary vector of length $|X|$ with non-zero entries for elements already crafted by the agent within the current episode (we refer to this as inventory $i$) and another binary vector of length $|X|$ where an index is non-zero if it is currently selected by the agent (we refer to this as current $c$). An agent begins with an inventory having non-zero element only for the initial set $\mathcal{X}_0$ and an all-zero selection. With the first action $a_0$, the selected item becomes non-zero in the selection. With the second action, $a_1$, we check if the combination $(a_1, c_0)$ is valid under the recipe book and, if so, return the newly crafted element (corresponding entry in $i$ becomes non-zero) and the reward. This two-step procedure continues until the end of the episode.

# B  SUMMARY OF RELATED WORKS

In this appendix we provide a non-comprehensive summary of the literature on the topic of the effect of social network topology on collective search in Table 1, where our objective is to highlight similarities and differences within and across the fields of cognitive science and DRL.

# C  PSEUDOCODE OF SAPIENS

We present the pseudocode of our proposed algorithm SAPIENS in Algorithm 1. SAPIENS works similarly to an off-policy reinforcement learning algorithm, with the difference that, after each episode, an experience sharing phase takes place between agents that belong in the same group.

**Algorithm 1** SAPIENS (Structuring multi-Agent toPology for Innovation through ExperieNce Sharing)

1: **Input:** $\mathcal{G}, connectivity, R, p_s, LS$
2: $\mathcal{G}$.initializeGraph(connecticity)
3: $\mathcal{I}$.initializeAgent()                                                  ▷ Initialize agents
4: **for** $i \in \mathcal{I}$ **do**
5:     I.neighbors= I.formNeighborhood($\mathcal{G}$)                    ▷ Inform agent about its neighbors
6:     I.env = initEnv(R)   ▷ Create agent's own copy of the environment based on the recipe book
7: **end for**
8: **while** training not done **do**
9:     **for** $i \in \mathcal{I}$ **do**                                                  ▷ Loop through each agent
10:        **while** episode not done **do**
11:            $a$ = i.policy()                                              ▷ Choose action
12:            $r, s_{new}$ = env.step(a)
13:            i.B.insert($[s, r, a, s_{new}]$)
14:        **end while**
15:        $\epsilon$ =random()
16:        **if** $\epsilon < p_s$ **then**                                      ▷ Share with probability $p_s$
17:            **for** $j \in$ i.neighbors **do**
18:                j.B.add(i.B.sample(L))              ▷ Sample random set of experiences of length $L$
19:            **end for**
20:        **end if**
        i.train()                                                  ▷ Train agent
21:     **end for**
22: **end while**

## D    ANALYSIS OF DYNAMIC NETWORK TOPOLOGIES

In the main paper we presented results for a single type of dynamic topolgoy. Here we present another type and analyze how they both behave for different values of their hyper-parameters. The two dynamic topologies are:

- Inspired by graphs employed in human laboratory studies (Derex & Boyd, 2016), we designed graphs where the macro structure of the graph is constant but agents can randomly change their position. In particular, we divide a group of agents into sub-groups of two agents and, at the end of each episode, move an agent to another group with a probability $p_v$ for a duration of $T_v$ episodes (for a visualization see Figure 3). To reduce the complexity of the implementation, we assume that only one visit can take place at a time. In the main paper we employ $p_v = 0.01$ and $T_v = 10$ across conditions and present results with different values in Appendix D, where we refer to this topology as dynamic-Boyd.

- Human behavioral ecology emphasize the importance of periodic variation in human social networks encountered throughout our evolutionary trajectory Wiessner (2014); Dunbar (2014). Due to ecological constraints human groups oscillate between phases of high and low connectivity: low-connectivity phases arise when individuals need to individually collected resources (e.g. day-time hunting) while high-connectivity phases arise when humans are idle and "forced" to be in proximity with others (e.g. fireside chats). Although these high-connectivity phases do not bare a direct evolutionary advantage, they may have played an important role by creating the conditions for the evolution of human language and culture. Inspired by this hypothesis, we have designed dynamic graphs that oscillate between a fully-connected topology that lasts for $T_h$ episodes and a topology without sharing that lasts for $T_l$ episodes. We present results for various values of $T_h$ and $T_d$ of this topology in Appendix D, where we refer to this topology as dynamic-periodic.

In Figure 8, we observe the % of group success ($S^{\mathcal{G}}$) with the dynamic-Boyd topology for different probabilities of visit ($p_v$) and visit duration $T_v$ (the sub-group size is 2 in all cases). We note that, due to our implementation choice that a visit can take place only if no other agent is currently on a visit, the visit duration also affects the mixing of the group: longer visits mean that fewer visits will

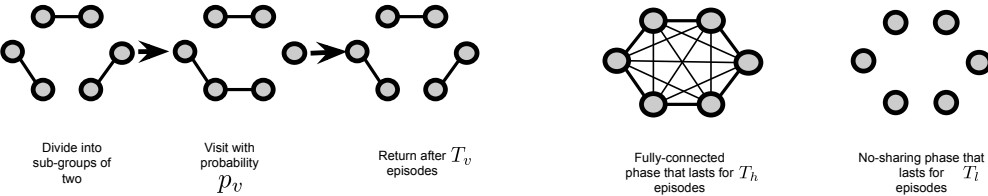

Figure 7: Two types of dynamic topologies: (Left) in the dynamic-Boyd topology the group is divided into sub-groups of two agents and a visit takes place with probability $p_v$ and lasts $T_v$ episodes (Right) In the dynamic-periodic topology the graph oscillates between a phase with a fully-connected topology that lasts for $T_h$ episodes to a phase without sharing that lasts for $T_l$ episodes.

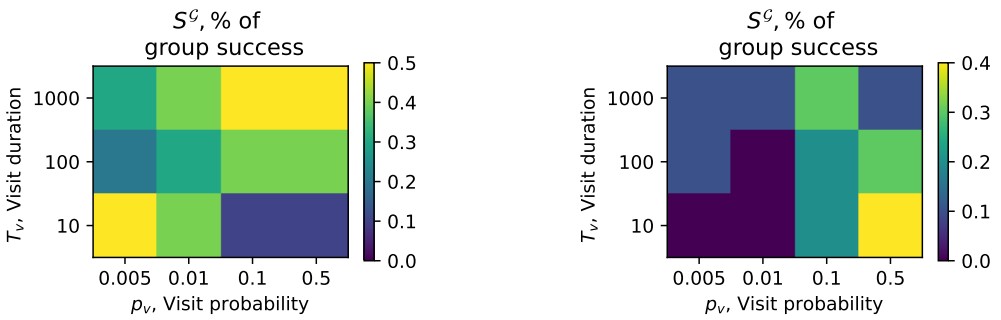

Figure 8: Examining the sensitivity of the dynamic-Boyd topology to its hyper-parameters: % of group success ($S^{\mathcal{G}}$) for the merging-paths task (left) and the best-of-ten paths task (right).

take place in total. **In the merging paths task (left), two hyper-parameter settings have a clear effect: (i) short visits with of high probability lead to bad performance. As such settings lead to a quick mixing of the population, they lead to convergence to the local optimum (ii) long visits with high probability work well. Due to the high visit probability, this setting effectively leads to topology where exactly one agent is always on a long visit. Thus, it ensures that sub-groups stay isolated for at least 1000 episodes, after which inter sub-group sharing needs to takes place to ensure that the sub-groups can progress quickly. In the best-of-ten paths task (right), this structure has a clear optimal hyper-parameterization: short visits with high probability are preferred, which maximizes the mixing of the group and makes early exploration more effective.**

In Figure 9, we observe the % of group success ($S^{\mathcal{G}}$) of the dynamic-periodic topology for various values of $T_h$ and $T_l$. **In the merging paths task (left of Figure 9) medium values for the period of both phases works best, while there is some success when the low connectivity phase lasts long ($T_l = 1000$). In the best-of-ten paths task (rightof Figure 9), we observe the same medium values for the period of both phases work best: thus both the absolute value and their ratio is important to ensure that exploration is efficient. The optimal configuration is the same between the two tasks ($T_l = 100, T_h = 10$), which is a good indication of the robustness of this structure.**

# E    EMPIRICAL RESULTS

To ensure that all methods have the same number of samples, we assume that, for trials where a method did not find the optimal solution, and, hence, $T^+$ is undefined, $T^+$ is equal to the total number of timesteps the method was trained for, $T_{\text{train}}$. For each task, all methods have been trained for an equal duration of time: $T_{\text{train}} = 1e^6$ for the single path , $T_{\text{train}} = 7e^6$ for the merging paths task and $T_{\text{train}} = 2e^7$ for the best-of-ten paths task.

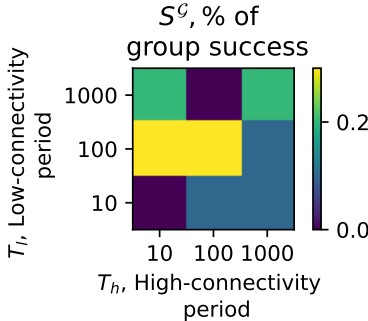 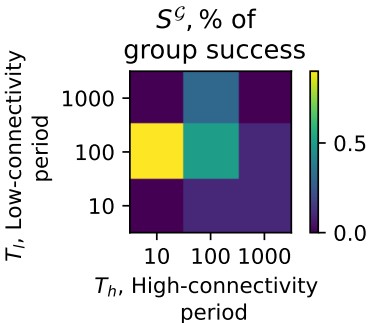

Figure 9: Examining the sensitivity of the dynamic-periodic topology to its hyper-parameters: % of group success ($S^{\mathcal{G}}$) for the merging-paths task (left) and the best-of-ten paths task (right).

We perform 20 independent trials for each task and method and visualize our proposed metrics with barplots and line plots of averages across trials with error bars indicating 95% confidence intervals. We test for statistical significance of our evaluation metrics separately for each task by applying ANOVA tests [8] to detect whether at least one method differs from the rest and, subsequently, employing the Tukey's range test [9] to detect which pairs of methods that differ significantly. We report the exact $p$ values of theses tests in the text and, when applicable, illustrate them in figures using a set of asterisks whose number indicates the significance level ($p <= 0.05$: *, $p <= 0.01$: **, $p <= 0.001$: ***, $p <= 0.0001$: **** ) [10].

We presented the major results of our evaluation of SAPIENS in Section 3. We now present additional information regarding the implementation of the different components (Appendix E.1), the values of all performance metrics and additional plots for experiments discussed in 3 (Appendix E.2), results on intra-group and inter-group alignment (AppendixE.3), results for groups of varying sizes (Appendix E.5) and results on various dynamic topologies (Appendix D)

### E.1 IMPLEMENTATION DETAILS

**Implementation of DQN**   We employ the same hyper-parameter for each DQN across all studied tasks and topologies: discount factor $\gamma = 0.9$, the Adam optimizer with learning rate $\alpha = 0.001$ (Kingma & Ba, 2014; Dunbar, 2014), $\epsilon$-greedy exploration with $\epsilon = 0.01$. We employ a feedforward network with two layers with 64 neurons each. We implemented SAPIENS by extending the DQN implementation in the stable-baselines3 framework.

**Implementation of A2C**   We used the stable-baselines3 implementation of A2C [11] and tuned the hyper-parameters: learning rate, number of steps, discount factor, the entropy coefficient and the value function coefficient. This gave us the best-performing values 0.001, 5, 0.99, 0.1 and 0.25, respectively, that we also employed in the other tasks.

**Implementation of Ape-X**   We used the ray implementation of Ape-X DQN [12] and tuned the hyper-parameters: learning rate, discount factor, replay buffer capacity and $\epsilon$-greedy exploration. This gave us the best-performing values in the single path task 0.001, 0.9, 5000 and 0.02, respectively.

---

[8]https://docs.scipy.org/doc/scipy/reference/generated/scipy.stats.f_oneway.html

[9]https://pypi.org/project/bioinfokit/0.3/

[10]https://www.graphpad.com/support/faq/what-is-the-meaning-of–or–or–in-reports-of-statistical-significance-from-prism-or-instat/

[11]https://stable-baselines3.readthedocs.io/en/master/modules/a2c.html

[12]https://docs.ray.io/en/latest/rllib/rllib-algorithms.html

| Topology | $R^+_\infty$ | $R^*_\infty$ | $T^+$ | $T^*$ | $T^>$ | $S$ | $V_{avg}$ | $C_{avg}$ |
|---|---|---|---|---|---|---|---|---|
| no-sharing | (0.92, 0.0036) | (1,0) | (236250, 33441) | (830000,0) | (600000, 0) | (1,0) | (0.038,0.002) | (0.697, 0.0354) |
| dynamic | (1,0) | (1,0) | (237222, 53885) | (346666,122041) | (109444, 98067) | (1,0) | (0.027,0.01) | (0.885, 0.026) |
| fully-connected | (1,0) | (1,0) | (310666, 89240) | (362000, 98503) | (51333, 20655) | (1,0) | (0.052, 0.027) | (0.891,0.034) |
| ring | (1,0) | (1,0) | (235333, 70190) | (305333, 78818) | (70000, 22038) | (1,0) | (0.038,0.0026) | (0.697, 0.0354) |
| small-world | (1,0) | (1,0) | (253333, 63320) | (302666, 74110) | (49333, 31274) | (1,0) | (0.029, 0.013) | (0.912, 0.0267) |
| single | (0.92, 0.163) | (0.927, 0.163) | (64750, 266145) | (64750, 266145) | (0,0) | (0.2, 0.41) | (0.015, 0.013) a non-co | (1,0) |
| A2C | (1,0) | (1,0) | (36200, 16450) | (36200, 16450) | (0,0) | (1,0) | (0,0) | (1,0) |
| Ape-X | (0.93, 0.18) | (0.93, 0.18) | (270941, 102445) | (270941, 102445) | (0,0) | (0.15, 0.366) | (0.015, 0.022) | (1,0) |

Table 2: Evaluation metrics for the single-path task in the form (mean of metrics, standard deviation of metric)

| Topology | $R^+_\infty$ | $R^*_\infty$ | $T^+$ | $T^*$ | $T^>$ | $S$ | $C_{avg}$ | $V_{avg}$ |
|---|---|---|---|---|---|---|---|---|
| no-sharing | (0.657, 0.037) | (0.838,0).14 | (5334000, 2311945) | (7000000,2311945) | (7000000, 0) | (0.4,0.51) | (0.597, 0.06) | (0.0089,0.0021) |
| dynamic | (0.7,0.04) | (0.9,0.13) | (4716500,222965) | (7000000,0) | (7000000, 0) | (0.75,0.48) | (0.597, 0.0059) | (0.005, 0.0016) |
| fully-connected | (0.5349,0.085) | (0.58, 0.04) | (7000000,0) | (7000000, 0) | (7000000, 0) | (0,0) | (0.597,0.0051) | (0.0764, 0.0044) |
| ring | (0.661,0.135) | (0.72, 0.15) | (5892000, 2288393) | (7000000, 0) | (7000000, 0) | (0.2,0.41) | (0.595, 0.0051) | (0.0149,0.021) |
| small-world | (0.639,0.091) | (0.774, 0.173) | (5998000, 1699076) | (7000000,0) | (7000000,0) | (0.3, 0.483) | (0.596,0.0065) | (0.06775,0.0328) |
| single | (0.758, 0.187) | (0.758, 0.187) | (5235000, 2385948) | (5235000, 2385948) | (0,0) | (0.3, 0.47) | (1,0) | (0.0063, 0.0063) |
| A2C | (0.269,0).2 | (0.269, 0.2) | (7000000, 0) | (7000000, 0) | (0,0) | (0,0) | (1,0) | (0.013, 0.038) |
| Ape-X | (0.573, 0.31) | (0.573, 0.31) | (6656900, 1534389 ) | (26656900, 1534389 ) | (0,0) | (0.05, 0.223) | (1,0) | (0.054,0.157) |

Table 3: Evaluation metrics for the merging-paths task in the form (mean of metrics, standard deviation of metric)

**Implementation of graphs used as social network structures** We construct small-worlds using the Watts–Strogatz model (`watts_strogatz_graph` method of the networkx package [13]). This model first builds a ring lattice where each node has $n$ neighbors and then rewires an edge with probability $\beta$. Compared to other techniques used in previous works studying the effect of topology Mason et al. (2008), this way of constructing small-worlds ensures that the average path lengths is short and clustering is high. These two properties are what differentiates small-worlds from random (short average path length and small clustering) and regular (long average path length and high clustering) graphs. We employ $n = 4$ and $\beta = 0.2$ in our experiments, which we empirically found to lead to good values of average path length and clustering.

We have described the generation process of dynamic topologies in Appendix D. In the main paper we employ the dynamic-Boyd topology with $T_v = 10$ and $p_v = 0.001$ across tasks. These parameters have been tuned for the merging-paths task.

## E.2 OVERALL COMPARISON

Tables 2, 3 and 4 contain the values of all metrics discussed in Section 2.4 for the single path, merging paths and best-of-ten paths, respectively. We denote values computed after convergence of the group with underscore $\infty$ and values averaged over all training steps with underscore $avg$ (note that we use ¯ over variables to denote averaging over agents in a single training step). Cells with a dash (-) indicate that we could not compute the corresponding metrics because a group failed to find a solution in all trials. We also provide the plots of volatility and average diversity for the merging paths and best-of-10 paths task (that were not included in Figure 5) due to page limit constraints.

Figure 10 presents the reward curves for all methods in the single path, merging paths and best-of-ten paths tasks respectively. Specifically, we plot the maximum reward of the group at training step $t$ ($R^+_t$).

---

[13] https://networkx.org/

| Topology | $R^+_\infty$ | $R^*_\infty$ | $T^+$ | $T^*$ | $T^>$ | $S$ | $C_{avg}$ | $V_{avg}$ |
|---|---|---|---|---|---|---|---|---|
| no-sharing | (0.2124,0.036) | (0.446, 0.131) | (20000000, 0) | (20000000,0) | (20000000, 0) | (0,0) | (0.239, 0.005) | (0.007, 0.0021) |
| dynamic | (0.5141,0.323) | (0775, 0.32) | (13616000, 5441395) | (20000000,0) | (20000000, 0) | (0.6,0.51) | (0.242,0.0078) | (0.04,0.0223) |
| fully-connected | (0.1615,0.09) | (0.1819, 0.1013) | (20000000, 0) | (20000000,0) | (20000000,0) | (0,0) | (0.238,0.0053) | (0.007,0.003) |
| ring | (0.2319,0.3045) | (0.275,0.332) | (18781000, 3854816) | (18826000, 3712513) | (18045000, 6182252) | (0.1,0.31) | (0.237,0.004) | (0.047,0.019) |
| small-world | (0.198,0.281) | (0.216,0.275) | (18706000, 4091987) | (18746000, 3965496) | (18040000, 6198064) | (0.1,0.316) | (0.234,0.007) | (0.018,0.0049) |
| single | (0.178, 0.067) | (0.1785,0.0676) | (20000000, 0) | (20000000,0) | (0,0) | (0, 01) | (1,0) | (0.006,0.0031) |
| A2C | (0.1285,0.19) | (0.1285,0.19) | (20000000, 0) | (20000000,0) | (0,0) | (1,0) | (1,0) | (0.3244,0.35) |
| Ape-X | (0.481, 0.213) | (0.482, 0.213) | (20000000, 0) | (20000000,0) | (0,0) | (0.9, 0.316) | (1,0) | (0.018,0.009) |

Table 4: Evaluation metrics for the best-of-ten paths task in the form (mean of metrics, standard deviation of metric)

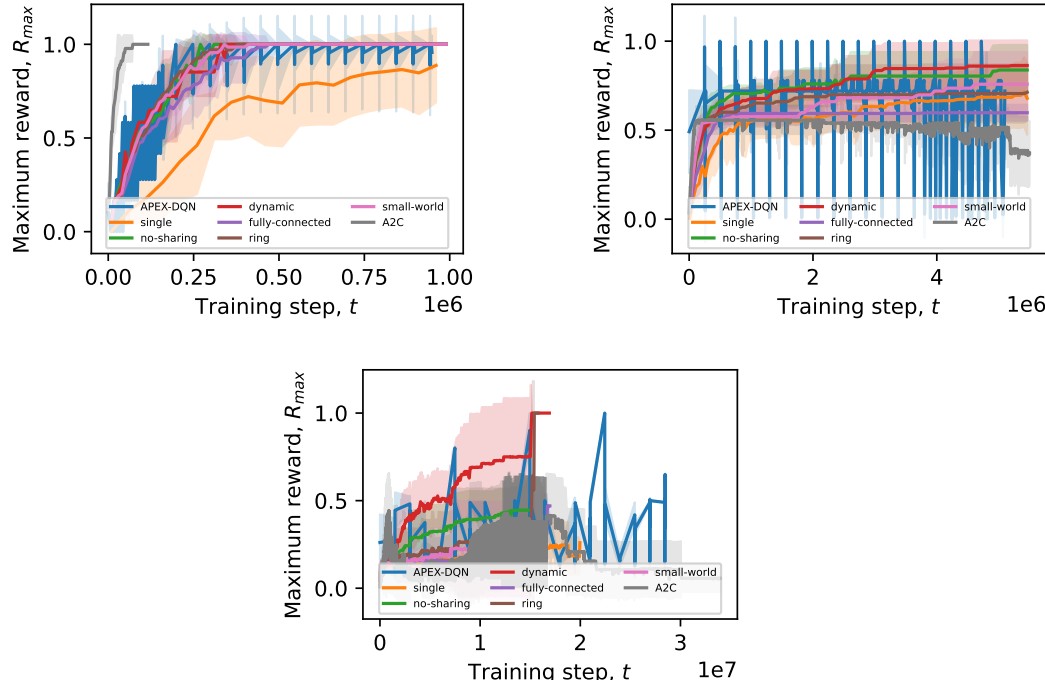

Figure 10: Maximum reward of the group at training step $t$ ($R_t^+$) in the (left) single path task (middle) merging paths task (right) best-of-ten paths task

### E.3 MEASURING INTER-GROUP AND INTRA-GROUP ALIGNMENT

We have so far captures the agreement between agents in a group through the behavioral metric of conformity. Here, we present a mnemonic metric for agreement, which we term alignment. Alignment is a complementary metric to the diversity ($D_t^k$) and group diversity ($D_t^{\mathcal{G}}$) metrics, that aims at capturing the effect of experience sharing on the replay buffers in a group. We propose a definition of alignment within a single group (intra-group alignment $A_t^{\mathcal{G}}$) and a definition of alignment between two different groups ($A_t^{\mathcal{G}_j, \mathcal{G}_j}$). Such metrics of mnemonic convergence have been linked to social network topology (Coman et al., 2016) and, as we show here, they can prove useful in analyzing groups of reinforcement learning agents.

Specifically: (i) $A_t^{\mathcal{G}}$ is the intra-group alignment. This metric captures the similarity in terms of content between the replay buffers of agents belonging to the same group. To compute this we compute the size of the common subset of experiences for each pair of agents and, then, average over all these pairs, normalizing in [0,1]. (ii) inter-group alignment $A_t^{\mathcal{G}_j, \mathcal{G}_j}$ is a similar notion of alignment but employed between different groups (e.g. how different is a group of fully-connected and a dynamic group of agents in terms of the content of their group replay buffers). To compute it we concatenate all replay buffers of a group into a single one and then compute the size of the common subset of the two replay buffers.

Figure 12 presents intra-group alignment in the three tasks. **We observe that, in all tasks, intra-group alignment increases with connectivity and that it reduces when the agents enter the exploitation phase. Thus, intra-group alignment can prove useful in characterizing the exploration behavior of a group.** In Figure 13, we present the inter-group alignment in the single path, merging paths and best-of-ten paths tasks. We observe that the topologies do not differ significantly in the single path task. **In the merging task, we observe that inter-group alignment is lower during the exploration phase, compared to other tasks, and that the small-world is the slowest to align with all other structures. Perhaps this explains why this topology finds the optimal solution with the least probability: by propagating information quickly, the group early on**

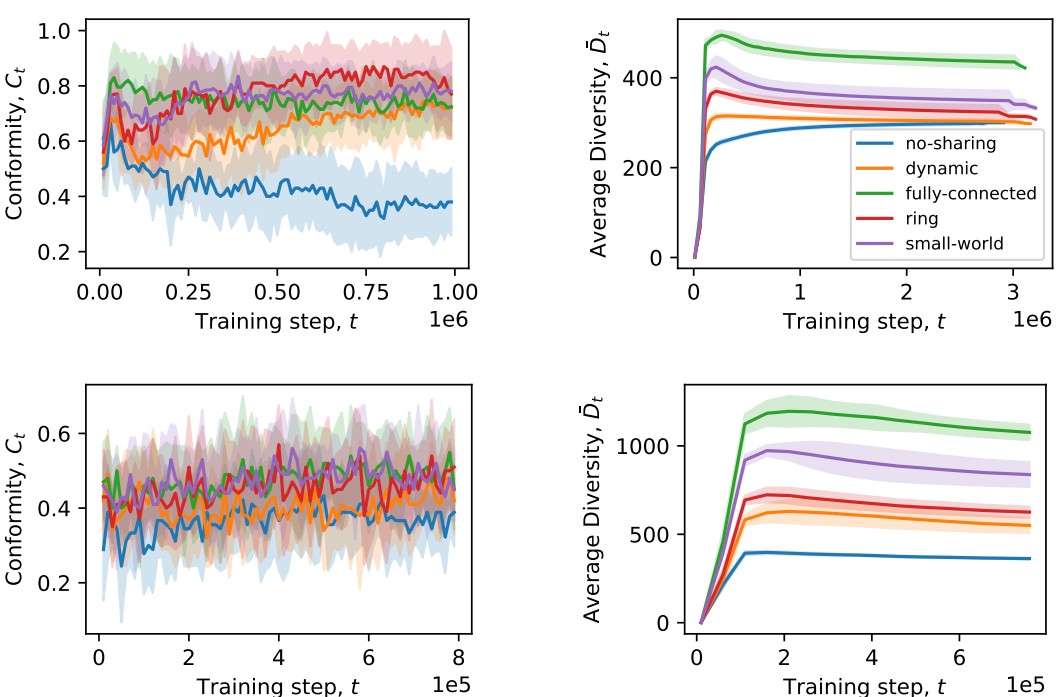

Figure 11: Analyzing group behavior in the merging paths task (top row) and best-of-10 paths task (bottom row). (left) Conformity $C_t$ is a behavioral metric that denotes the percentage of agents in a group that followed the same trajectory in a given evaluation trial (right) Average Diversity $\bar{D}_t$ is a mnemonic metric that denotes the number of unique experiences in the replay buffer of an agent, averaged over all agents.

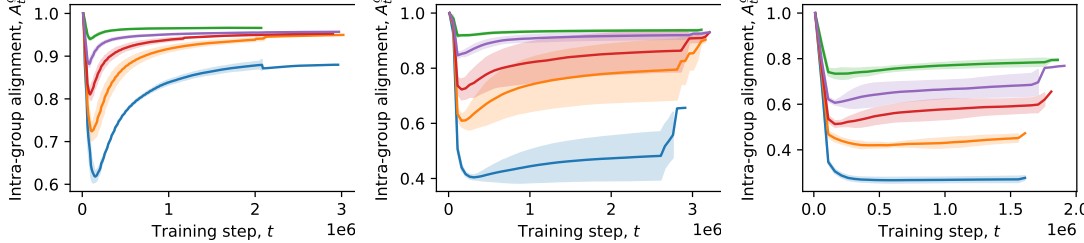

Figure 12: Intra-group alignment $A_t^{\mathcal{G}}$ in the single path task (left), merging paths task (middle) and best-of-ten paths task (right)

**converges to the local optimum in this task. In the best-of-ten task, the no-sharing setting has the smallest alignment with all other structures. This reinforces our main conclusion in this work: experience sharing affects individuals and different topologies do so in different ways.**

### E.4 ROBUSTNESS TO LEARNING HYPER-PARAMETERS

In Figure 14 we present how the performance of SAPIENS varies for different values of the learning hyperparameters learning rate and disocunt factor in the single path task under a fully-connected and a dynamic topolgoy, as well as the *no-sharing* condition. We observe that, although convergence to the optimal solution is not always achieved, the dynamic topology is at least as effective as the others either in terms of convergence rate and/or final performance in all conditions.

### E.5 EFFECT OF GROUP SIZE

We here examine the effect of the group size for all social network structures in the merging-paths and best-of-ten paths task. To visualize the progression of a group on the paths of the different tasks, we focus on specific elements in the tasks: (i) ($[A_8, B_8, C_2]$ in the merging-paths task. The first two correspond to reaching the end of the paths corresponding to the two local optima. To reduce the computational complexity of experiments, we do not study the last element of the optimal path ($C_4$), but focus on $C_2$ instead. This is sufficient to detect whether a group has discovered the optimum path. Here, we observe that the fully-connected topology fails to find the optimal path regardless of its size (with a small success probability for $N = 10$). We observe that the ability of the ring , small-world and dynamict topologies to avoid the local optima improves with the group size (ii) $[B_4, A_2, E_2]$ in the best-of-ten tasks. $B_4$ is the fourth element on the optimal path (again we do not study the last element to reduce complexity). To avoid cluttering the visualization we only present two of the nine sub-optimal paths. In this task, we again observe that the fully-connected network fails to discover the optimal task. Among all structures and group sizes, the large dynamic network performs best, while the performance of ring and small-world is also best for $N = 50$. We observe that small networks sizes ($N = 2, N = 6$) are slower at exploring (we can see that as they rarely find the second element of the sub-optimal paths, which is required to conclude that path $B$ is the optimal choice).

Overall, **this scaling analysis indicates that increasing the group size in a fully-connected topology will not improve performance, while benefits are expected for low-connectivity structures, particularly for the dynamic topology.** We believe that this observation is crucial. In studies of groups of both human and artificial agents, we often encounter the conviction that, larger groups perform better and that size is a more important determinant than connectivity, the latter justifying why connectivity is often ignored Kline & Boyd (2010); Horgan et al. (2018); Mnih et al. (2016); Schmitt et al. (2019); Nair et al. (2015). Our results here point to the contrary.

### E.6 PRIORITIZED EXPERIENCE SHARING

We now examine how sharing prioritized experiences instead of randomly sampled ones affects the performance of SAPIENS. In Figure 16 we repeat the same experiment with Figure 4, with the difference that all methods compute priorities, which they employ both for implementing a prioritized

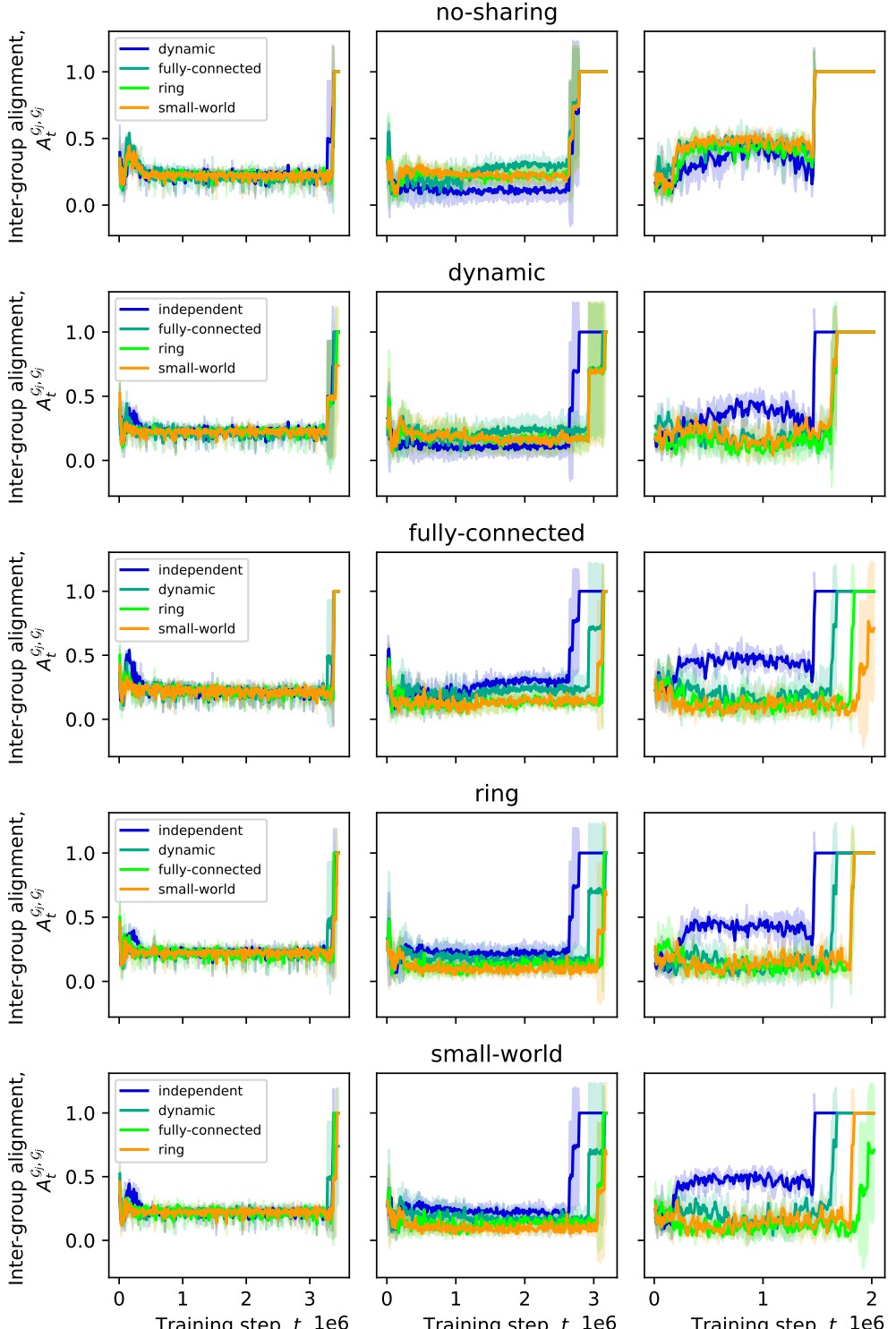

Figure 13: Inter-group alignment $A_t^{\mathcal{G}_j,\mathcal{G}_j}$ in the single path task (left), merging paths task (middle) and best-of-ten paths task (right). In each row we compare one topology with all the rest.

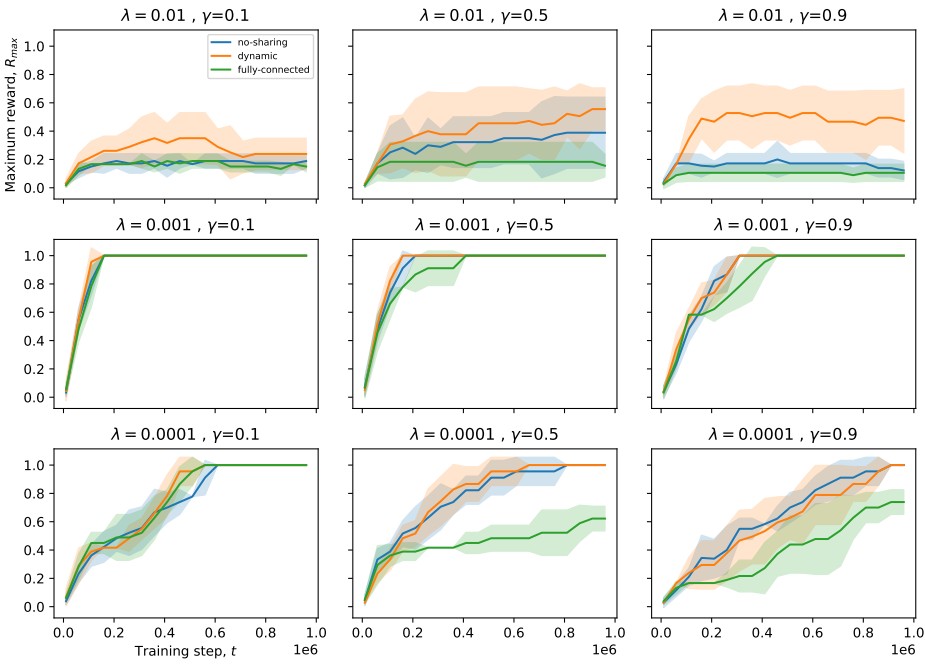

Figure 14: Varying the learning hyper-parameters learning rate ($\lambda$) and discount factor ($\gamma$) in different social network topologies in the single path task. )

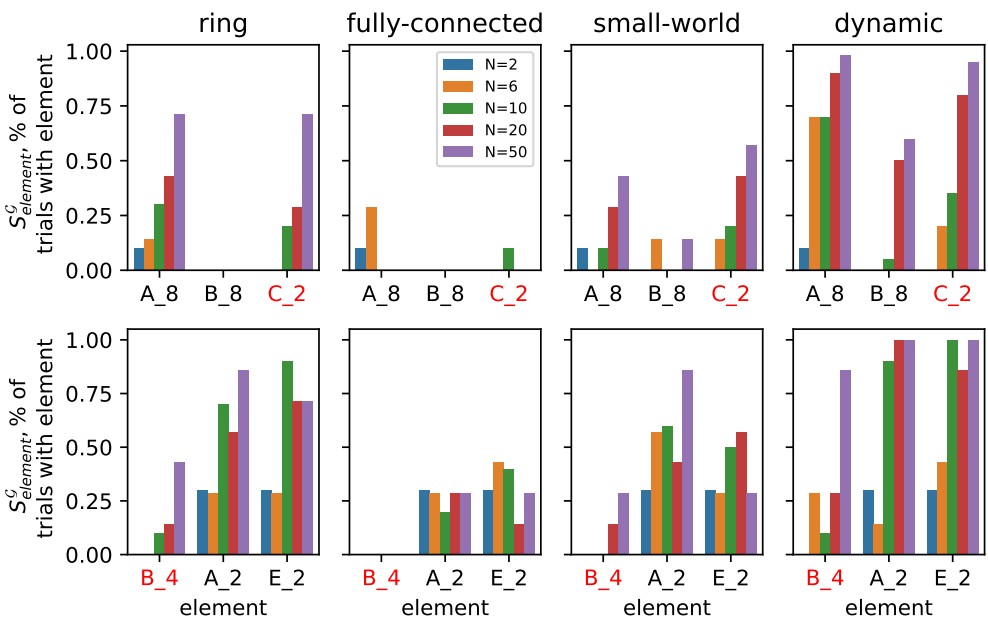

Figure 15: Scaling of different social network structures in the merging paths (top row) and best-of-ten paths tasks (bottom row). We highlight the element belonging to the optimal path in red.

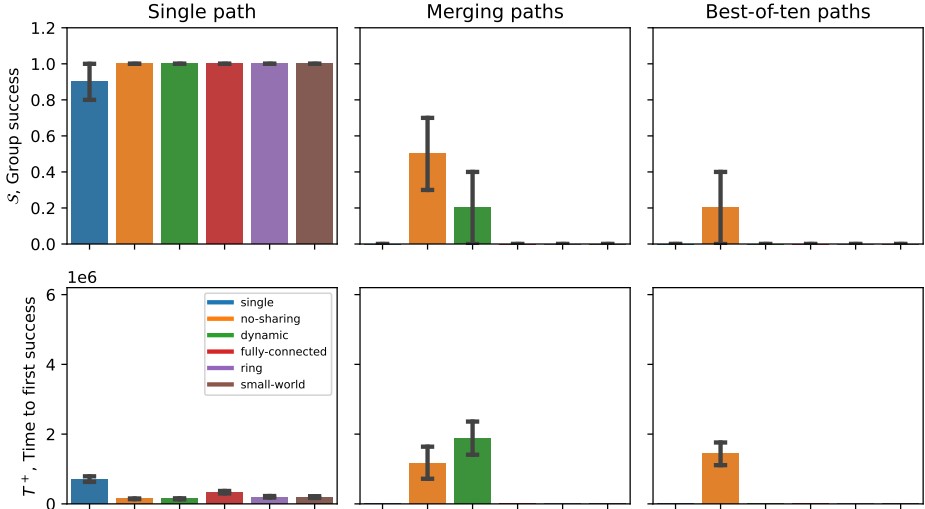

Figure 16: Examining the effect of prioritization in experience sharing. For more details about the setup, we refer the reader to Figure 4

replay buffer and sharing experiences by sampling them in proportion to their priorities. As we see, using priorities negatively impacts experience sharing, while it helps speed up the performance of the single agent in the single path task. This behavior has been observed in previous works Souza et al. (2019) and can be attributed to the fact that the priorities of the sender do not necessarily agree with the priorities of the receiver and, therefore, destabilize learning.

### E.7 ADDITIONAL TEST-BED: THE DECEPTIVE COINS GAME

Deceptive games are grid-world tasks introduced to test the ability of deep RL agents to avoid local optima. (Bontrager et al., 2019). Here, we perform preliminary experiments with our own JAX-based implementation of one of the games: the first difficulty level of the deceptive coins game (see Figure 17 for an illustration). Here, the agent can navigate in the grid-world during an episode and collect diamonds, which give a unit of reward. The game finishes once the agent reaches the fire, which offers an additional reward, or when a timeout of 14 time steps is reached. There are two possible paths the agent can follow: moving left and reaching the fire will give a reward of two while moving right and reaching the fire will give a reward of five. The second path is more rewarding but is harder to complete because, once an agent discovers the easier-to-find diamond on the left, it is deceived into following the left path. Once an agent commits on a path (reaches the edge of the grid-world) a barrier is raised so that the agent cannot go back within that episode.

We now examine the performance of SAPIENS under different social network structures (fully-connected, small-world, ring, dynamic), as well as the no-sharing, A2C and Ape-X baselines for three group sizes: 6 , 10 and 20 agents. We present the reward plots for the 3 sizes in Figures 18, 19 and 20, respectively, and present an overall comparison in Figure 21 (equivalent to Figure 4 for the Wordcraft tasks).

We observe that all conditions found either the local or the global optimum and that : a) A2C fails for all network sizes. This behavior has been observed in previous works (Bontrager et al., 2019) and can be attributed to the fact that policy-gradient methods are more susceptible to local minima b) no-sharing gets stuck in the local optimum in half of the trials when the group size is small. Increasing the group size increases the probability that at least one agent in the group will escape the local minima by $\epsilon$-greedy exploration c) partially connected structures find the global minima across network sizes d) fully-connected converges to the local optimum for the large group size, although the global optimum was discovered at the early exploration phase (see Figure 20). Thus, too much experience sharing is harmful e) Ape-X fails with high probability for all network sizes.

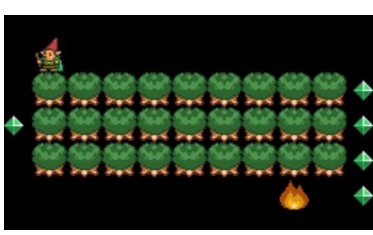

Figure 17: A screenshot of our implementation of the Deceptive Coins task. Collecting diamonds gives a positive reward and touching the fire terminates the game.

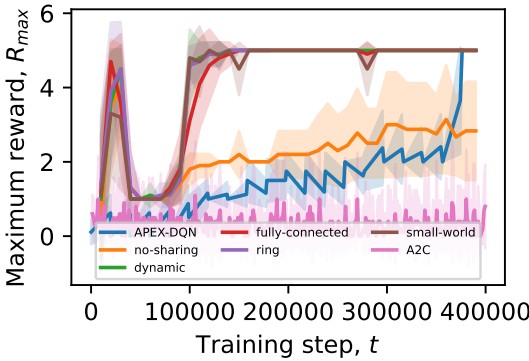

Figure 18: Performance for a group with 6 agents

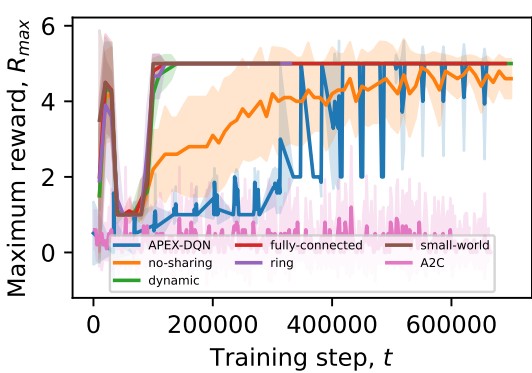

Figure 19: Performance for a group with 10 agents

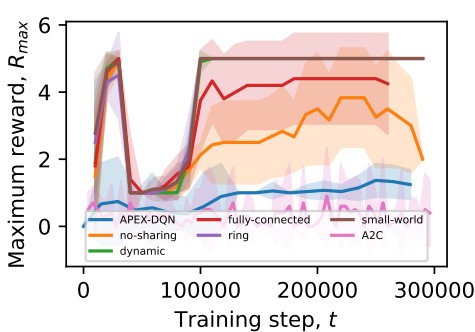

Figure 20: Performance for a group with 20 agents

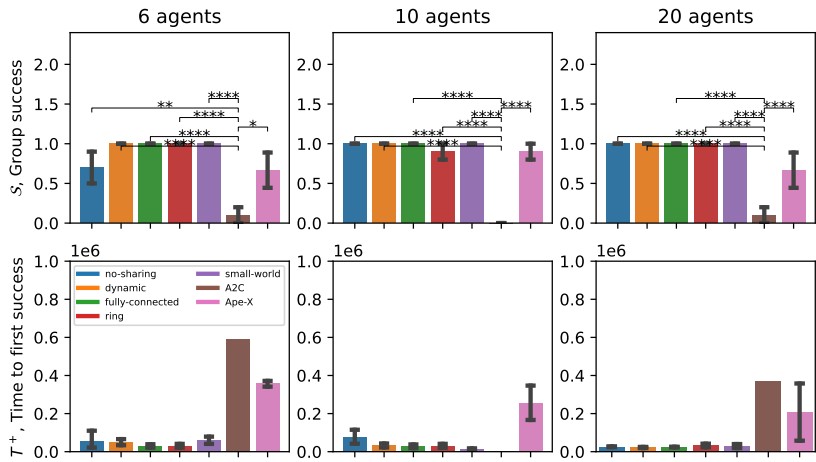

Figure 21: Overall performance comparison for a group with: 6 agents (first column), 10 agents (second column) and 20 agents (third column) task. We present two metrics: group success ($\mathcal{S}$) denotes whether at least one agent in the group found the optimal solution (top row) and $T^{+}$, Time to first success, is the number of training time steps required for this event (bottom row). Note that $T^{+}$ can be computed only for $\mathcal{S} > 0$ its error bars and significance tests can only be computer for $\mathcal{S} > 1$. We denote statistical significance levels with asterisks.)

In general, our conclusions in this task are consistent with what we observe in Wordcraft, in particular the merging paths task that has a similar deceptive nature.

### E.8 ROBUSTNESS TO AMOUNT OF SHARING ($p_s$ AND $L_s$)

In Section 2.3 we formulated SAPIENS and described two hyper-parameters: $p_s$ is the probability of sharing a batch of experience tuples at the end of an episode and $L_s$ is the length of this batch. Here, we test the robustness of SAPIENS to these two hyper-parameters, which both control the amount of shared information and, therefore, interact with hyper-parameters of the DQNs (in particular the learning rate) to control the rate at which information is shared to the rate of individual learning. Specifically, we evaluate the dynamic topology (with the same hyper-parameters employed in the main paper, i.e., visit duration $T_v = 10$ and probability of visit $p_v = 0.05$) and the fully-connected topology in the deceptive coins game (described in Appendix E.7) with 20 DQN agents.

In Figure 22 we present group success ($S$) averaged across trials for a parametric analysis over $L_s \in (1, 6, 36)$ and $p_s \in (0.35, 0.7, 1)$. We observe that the dynamic topology finds the optimal solution across conditions except for a small probability of failure for ($L_s = 1, p_s = 0.35$) and ($L_s = 1, p_s = 0.7$). These values correspond to low amounts of information sharing. In this case, the dynamic structure becomes more similar to a no-sharing structure: the amount of shared information is not enough to help the agents avoid local optima they fall into due to individual exploration. For the fully-connected topology we observe that performance degrades for high amounts of information (($L = 36, p_s = 0.35$), ($L = 36, p_s = 0.7$), ($L = 36, p_s = 1$)). This is in accordance with our expectation that fully-connected topologies lead to convergence to local optima. Interestingly, this structure performs well when $p_s = 1$ and $L_s \leq 6$. Thus, sharing more frequently is better than sharing longer batches: we hypothesize that this is because longer batches have more correlated data, making convergence to local optima more probable.

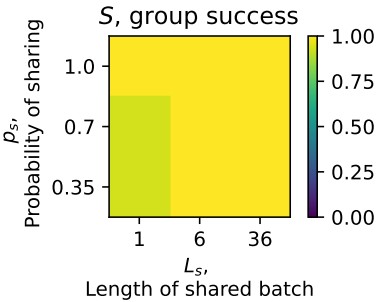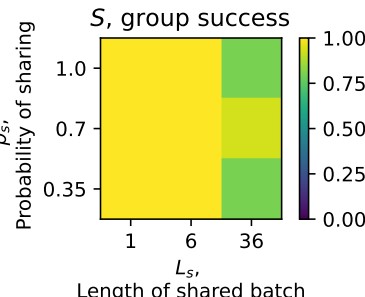

Figure 22: Robustness of group success $S$ to sharing hyper-parameters $p_s$ and $L_s$ for dynamic (left) and fully-connected (right)

