# OpenReview forum: "Social Network Structure Shapes Innovation: Experience-sharing in RL with SAPIENS"
_ICLR.cc/2023/Conference — Submitted to ICLR 2023_

### Official Review · Reviewer_GCRF · 2022-10-21

**Confidence:** 2
**Correctness:** 4
**Technical Novelty And Significance:** 3
**Empirical Novelty And Significance:** 3
**Recommendation:** 6

**Clarity, Quality, Novelty And Reproducibility:**

The paper is very clearly written but could use some improvements in some of its figures (see the weaknesses section). The quality of the paper and reproducibility are also high; there is code and it is clear how to reproduce the experiments in the paper based on the details contained in the paper and its appendices.

**Strength And Weaknesses:**

The paper is a generally strong paper. It is attacking a significant problem or interest to the community (i.e. how to do multi-agent better reinforcement learning), is well grounded in both network and RL literature and methods, and has some important contributions. I consider the insight into why certain network topologies work as well to be an especially important contribution, as it could guide future research. I also very much appreciate that the insights drawn in the paper are based on appropriate metrics, like the volatility and diversity metrics.

The weaknesses of the paper are few. First, the setup for the evaluation seems rather simple and contrived (discrete tasks and only a few path combinations). It doesn’t have any uncertainty related to task performance or much complexity in the task paths. However, the paper does address this in the limitations, and I don’t consider this as a serious issue since one has to start somewhere, and stating with a simple base-case is good science. Second, figure 4 is not very clear. Does the lack of bars indicate total task failure for a method, especially in the Best-of-Ten paths? Also, the significance brackets – do the endpoints of the bracket indicate a significant difference between the methods at the endpoints? And, what do the stars indicate (I presume the number of stars is the level of statistical significance)?


**Summary Of The Paper:**

The paper presents an investigation of how to network reinforcement learning models to accomplish symbolic innovation tasks. The paper analyzes several different, common network topologies as a means of sharing experiences between DQN learners as well as other existing methods that either share experiences or gradients between the learners. The paper considers three different types of innovation paths as well as a ‘deceptive task’ as part of its evaluation of the different sharing configurations. The paper also offers insights into why some network structures work (or do not work) for different tasks as well as some practical guidance for actual usage.

**Summary Of The Review:**

I believe this paper merits acceptance as it is a high-quality paper, with a significant contribution that would be of interest to the community. The work in the paper on how to combine multi-agent reinforcement learners is a significant problem. And the experimental regime used well supports the conclusions drawn in the paper.

--------After Reviewing Period--------------
Following the discussion period and consultation with the other reviewers and chair, I have revised my assessment and lowered my recommendation.

---

> ### Author Response · Authors · 2022-11-17
> **Overall reply to reviewer GCRF**
>
> We thank reviewer GCRF for finding the paper strong in terms of the research question asked, its grounding in the literature, its methodology and contributions and for mentioning some weaknesses, based on which, we improved the manuscript.
>
> We would like to note that we have revised our manuscript to address the comments raised by all reviewers, where we indicated changed text in blue. We encourage the reviewer to read the new version, but this is not necessary, as below we reply to their two points (we quote text that was added in the revised manuscript in italics):
>
> **Point 1: First, the setup for the evaluation seems rather simple and contrived (discrete tasks and only a few path combinations). It doesn’t have any uncertainty related to task performance or much complexity in the task paths. However, the paper does address this in the limitations, and I don’t consider this as a serious issue since one has to start somewhere, and stating with a simple base-case is good science.**
>
> We agree with the reviewer that our tasks are simple from a reinforcement learning perspective, but this was indeed done on purpose to draw clear conclusions.  We should note that we have performed preliminary experiments in a grid-world with a deceptive task that also entails the challenge of navigation and shown that our empirical observations about the differences between social network structures made in Wordcraft still apply (the experiments are presented in Appendix E.7 and mentioned in Section 3 ).  In the revised manuscript, we have updated our discussion of limitations, which in particular for RL mentions the following :
>
> *“ From an RL perspective, our study is limited in including experiments only in a few symbolic tasks and a simple navigation task; in the future we plan to extend to more complex environments like Crafter (Hafner, 2021).”*
>
> **Point 2: Second, figure 4 is not very clear. Does the lack of bars indicate total task failure for a method, especially in the Best-of-Ten paths? Also, the significance brackets – do the endpoints of the bracket indicate a significant difference between the methods at the endpoints? And, what do the stars indicate (I presume the number of stars is the level of statistical significance)?**
>
> Indeed, for group success $S$ (first row) a lack of bar means that all trials failed for a method. For the time to first success $T^+$ (second row) in our original manuscript we considered that failed trials had no $T^+$, but this gives rise to unequal samples for each method, which makes pairwise comparisons with the Tukey range test impossible. In the revised manuscript, we consider that failed trials have  equal $T^+$ to their total training time. We mention  this at the beginning of Appendix E  and refer to this appendix in Section 3:
>
> *“To ensure that all methods have the same number of samples, we assume that, for trials where a method did not find the optimal solution, and, hence, $T^+$ is undefined, $T^+$ is equal to the total number of timesteps the method was trained for, $T_{\text{train}}$. For each task, all methods have been trained for an equal duration of time: $T_{\text{train}}=1 e^6$ for the single path ,  $T_{\text{train}}=7 e^6$  for the merging paths task and $T_{\text{train}}=2 e^7$ for the best-of-ten paths task.”*
>
> The endpoints of the brackets indeed indicate pairwise comparisons and the asterisks denote the significance level. In the revised manuscript in section “3.1 Overall comparison” we mention:
>
> *“where we also indicate statistically significant pairwise comparisons with asterisks (more asterisks denote higher significance)”*
>
> And then provide more information in Appendix E:
>
> *“We perform 20 independent trials for each task and method and visualize our proposed metrics with barplots and line plots of averages across trials with error bars indicating $95\%$ confidence intervals. We test for statistical significance of our evaluation metrics separately for each task by applying ANOVA tests to detect whether at least one method differs from the rest and, subsequently, employing the Tukey's range test to detect which pairs of methods that differ significantly. We report the exact $p$ values of theses tests in the text and, when applicable, illustrate them in figures using a set of asterisks whose number indicates the significance level* ($p<=0.05$: *, $p<=0.01$: **, $p<=0.001$: ***, $p<=0.0001$: **** )

---

### Official Review · Reviewer_DXUd · 2022-10-24

**Confidence:** 4
**Correctness:** 2
**Technical Novelty And Significance:** 3
**Empirical Novelty And Significance:** 4
**Recommendation:** 5

**Clarity, Quality, Novelty And Reproducibility:**

Clarity & Quality (mostly):
- "ones(Hafner, 2021) b)" missing space
- "Watts & Strogatz (1998)." Citation should be in parentheses
- "vast spaces.."
- "and receives a reward that increases monotonically with levels" Missing period
- "spread between clusters" missing period
- The phrase "deceptive game" is misleading. The games do not have agency. Better to choose something like difficult, or non-obvious, or similar.
- Derex & Boyd (2016); Migliano & Vinicius (2022)
- "Specifically, methods ring, small-world, dynamic and fully-connected are instantiations of SAPIENS for different social network structures with 10 DQNs. " Confusing sentence. Are the italics important? Are these methods?
- When reporting p values, please report the full details that went into the calculation, including the means (and variances as appropriate), test statistic, degrees of freedom, the name of the test used, etc.
- "In contrast, fully-connected, A2C and Ape-X perform significantly worse. " This claim needs to be quantified.
- "In additional experiments we have also:" I find it unacceptable to introduce results without the methods and evidence. (Even if they are included in the Appendix.)
- "we hypothesize that it is due to the fact that priorities computed by an agent do not necessarily agree with the priorities of the other" What do you mean, why do you believe this, and how could it be tested?
- "We denote statistical significance levels with asterisks.)" This is not acceptable. Please state explicitly what the number of stars reflects. Also, given the enormous number of tests is there a preregistered evaluation plan? Have the analyses been corrected for the number of tests? Also, it is not clear that the trailing parenthesis is closing anything.
- "While it is not surprising that a single agent with epsilon-greedy exploration can efficiently solve this task, it is not clear at first glance why experience sharing harms performance. Notably this phenomenon has been observed in related works (Souza et al., 2019; Schmitt et al., 2019)." Is this intended to explain the surprise? It would be nice to provide some explanation for the observations.
- "Perhaps surprisingly, the fully-connected group does not have the highest conformity (Figure 5, Left). We therefore conclude that there is an upper threshold for connectivity, beyond which shared experiences destabilize learning." Why is the conclusion warranted? I do not follow the logic.
- "Finally, when looking at the group diversity in Figure 6, fully-connected ranks last and the dynamic topology ranks first. This is rather surprising: the" It would be nice to have experiments designed to test the observation.
- "Appendix ¡REF¿."

**Strength And Weaknesses:**

Strengths:
- The paper asks a big, interesting question, and does so in an interesting fashion. Specifically, the social nature of innovation is bedrock in the literature on human learning. Yet, we know relatively little about this question in the multi-agent RL setting. It is indeed an important, and interesting direction.
- The paper varies two simple, yet central features: the network topology and the nature of the problem. Building from the human learning (and other) literature, the test cases are informed by evidence and interesting.
- The method of sharing experience is quite natural for DQN learners, samples from the replay buffer, which is nicer than other work which shares gradients.
- There are a large number of experiments reported.

Weaknesses:
- "share experience tuples from their replay buffers" This is a substantial theoretical commitment regarding the form of knowledge and the nature of communication. It is important to discuss limitations that arise from these particular choices, especially with regard to generalization to human-AI and human-human innovation.
- "agents can share their experience by simply exchanging transitions from their respective replay buffers, without requiring ad-hoc mechanisms for copying the behaviors of other agents" It is not clear why exchanging transitions from reply buffers is less ad hoc.
- "which properties of experience sharing improve multi-level innovation" Is this question answered? I am unclear on what constitutes properties of experience in the context of the simulations. (I am assuming that is must be different from the network structures and tasks, as that is the first unresolved question.)
- Paper is ultimately more descriptive than conclusive. In principle, this need not be a weakness. However, there appear to be many opportunities to investigate further that were not taken. Also, the claims of the paper, especially related to implications outside of this model and these tasks, seem to require stronger causal understanding of why the phenomena occur.
- I am confused about what exactly SAPIENS is meant to be? Is it an experimental testbed? An optimization toolbox? A novel algorithm?
- " two metrics we propose, conformity and diversity of shared experience, can explain the success of different social network structures on different tasks" Is this intended as a correlational or causal claim. The language is ambiguous, but it evokes a causal interpretation. I did not see causal evidence however.
- "Here we test the hypothesis that the topology of experience sharing in a group of deep RL agents can shape its performance using our proposed algorithm SAPIENS." This hypothesis is first stated in the discussion. It doesn't seem to map onto the two open questions posed in the abstract?

**Summary Of The Paper:**

The paper investigates how network topology influences solving of different kinds of problems by DQN learners who exchange experience by sharing experience from their replay buffers. The paper specifically investigates network structures including  fully-connected, small world, ring and dynamic and tasks that vary in requiring a single innovation, multiple merged innovations, and broader search spaces. Performance is measured via a variety of metrics spanning performance, behavior, and "mnemonics". The results include some phenomena that might be predicted based on the existing human behavior literature, as well as several surprises. Overall, the authors conclude that both network topology and problem structure affect innovation in DQN agents.

**Summary Of The Review:**

On the one hand, the paper asks a timely and interesting question, and does so in an interesting way. However, I also believe the paper would benefit from extensive revision. There are a large number of minor typos, bad citations, and broken references, some but not all of which I note above. The claims of the paper are not totally clear; for example, see inconsistencies between the open questions and the hypothesis (which appears at the end). The statistical analyses are incompletely documented and are questionable. I was confused about what even is SAPIENS? (Does it need to be named?) Ultimately, the paper's claims appear to be primarily descriptive in nature, but I believe the authors would like them to be stronger (from the grand scope they invoke). Like many big ideas papers, the paper suffers for not having a clear enough focus about what precisely is the contribution and marshalling the evidence in a clear, compelling, and organized fashion to support that contribution. I believe this could be an excellent and pathbreaking paper, given substantial work to focus, edit, revise, and clarify.

---

> ### Author Response · Authors · 2022-11-17
> **General reply to reviewer DXUd**
>
> We thank reviewer DXUd for their constructive review and are happy they found the idea important and interesting and the number of experiments large.
>
> We have revised our manuscript, highlighting changed text in blue. The following changes will be of particular interest for reviewer DXUd:
>
> * We have done major changes in “3. Results”. In particular, we have rewritten our discussion of results to improve its quality by reporting statistically significant results and properly explaining how we used our evaluation metrics to derive our conclusions. We also provide more information about how we performed statistical tests in Appendix E
> * We have updated tables 2,3,4 in Appendix E.1 to include both means and standard deviations of all evaluation metrics in the three tasks. This appendix includes additional information about statistical tests (including what the asterisks refer to)
> * In Section “5. Discussion and future work” we explain the contributions of our empirical study to the two fields. We have removed the previous paragraphs that both reviewers Gs4s and GCRF found too broad. We also added a paragraph on limitations
> * We have revised  the abstract to improve its clarity.
>
>
> In follow-up posts we reply to each comment raised by reviewer DXUd separately, explaining which changes it led to. We quote all changes in italics so that the reviewer does not need to go back-and-forth, except for Section 3, which we kindly ask the reviewer to read from the new draft as it has significantly rewritten due to changes in the statistical analysis methods.

---

> ### Author Response · Authors · 2022-11-17
> **Reply to Point1: "share experience tuples ... human-human innovation" and Point 2: "agents can share ... ad hoc"**
>
> We agree with the reviewer that, when employing computational models to reproduce human behaviors, one needs to emphasize how artificial and human agents differ in the ways they learn and communicate. We would like to note however that, compared to previous computational studies where an agent maintained a belief (a value between 0 and 1) and adopted the belief of the majority in its neighborhood, our computational model is more similar to the experimental protocol usually employed in human studies. Comparing it to the study by Derex and Boyd, 2015, “Sharing experience tuples from their replay buffers” is similar to observing what others are doing and keeping it in one’s memory. Then, the policy is responsible for deciding how these experiences are used, instead of pre-assuming a copying mechanism (like the majority rule). This is what we mean with this sentence in the revised Section “1. Introduction”, where we clarify that by “less ad hoc” we meant in relation to previous computational models:
>
> *"agents can share their experience by simply exchanging transitions from their respective replay
> buffers, without requiring ad-hoc mechanisms for copying the behaviors of other agents, such as the
> majority rule (Lazer & Friedman, 2007; Cantor et al., 2021)"*
>
> We would also like to note that our revised draft does not contain the paragraph about “generalization to human-AI and human-human innovation”. As we received another comment by reviewer DXUd on “especially related to implications outside of this model and these tasks, seem to require stronger causal understanding of why the phenomena occur.” (see Point 4) and two comments by reviewer Gs4s about “explaining how we contribute to psychology” (see our reply with title [“Reply to Point 1: “[major] The manuscript needs…”](https://openreview.net/forum?id=BO5_Lm7iD_&noteId=xJR7prWz6P)) and “overstretching our claims in the conclusion” (see our reply with title [“Reply  to Point 3: “major” The manuscript overstretches…. ”](https://openreview.net/forum?id=BO5_Lm7iD_&noteId=lDKmXuAWjEL)), we have replaced the previous two paragraphs with a new paragraph that we quote here:
>
> *"We hope that our work will contribute to the fields of cognitive science and DRL in multiple ways.
> First, our empirical observations in the single path and best-of-ten-path tasks provide concrete hy-
> potheses for future experiments studying human innovation, which has so far been studied only in
> a task that inspired our merging-paths task (Derex & Boyd, 2016). By continuing the dialogue that
> has been initiated between human and computational studies (Fang et al., 2010; Lazer & Friedman,
> 2007; Cantor et al., 2021) to include DRL methods, we believe that cognitive science will benefit
> from tools that, as we show here, can learn in realistic problem set-ups and can be analyzed not
> just in terms of their behavior, but also in terms of their memories. Second, we hope that studies
> in distributed RL will extend their evaluation methodology by analyzing not just rewards, but also
> behavioral and mnemonic metrics such as diversity, conformity and volatility that, as we show here,
> correlate with success. Aside this, the effect of social network structure in distributed RL can be
> extended beyond evolutionary strategies (Adjodah et al., 2019) and beyond our current instantiation
> of SAPIENS, by considering other off-policy algorithms than DQNs and other types of information
> sharing. Finally, considering the effectiveness of the dynamic topologies observed in this study,
> we envision future works that investigate more types of them, as well as meta-learning or online-
> adaptation algorithms where the social network structure is optimized for a desired objective."*
>
> Even taking into account the above changes, we agree with the reviewer that we need to mention the limitation of this form of experience sharing. For this, we found the discussion in (Mason et al, 2008) about normative versus informational influence in humans, where the former appears in intellective tasks (eg mathematical optimization) while the latter appears in judgemental tasks (eg what is the best ice-cream flavor). Our agents are not normative (they do not desire social approval) and hence could not reproduce human studies in normative tasks. Even for intellective tasks, differences in the communication and decision-making mechanism  can lead to mismatches."*
>
> Based on this, we have added the following paragraph in the revised draft on limitations:
>
> *"When adopting RL algorithms as computational models for replicating experiments with humans,
> one needs to acknowledge that their communication and decision-making mechanisms may not
> faithfully replicate the ones used by humans. One notable difference is that humans may exhibit
> normative behavior, adopting information not for its utility in the task but for social approval (Mason
> et al., 2008)"*

---

> ### Author Response · Authors · 2022-11-17
> **Reply to Point 3: "which properties .. question)" and Point 7 "Here we test ... abstract?"**
>
> We have group points 3 and 7 together as they both refer to the abstract misstating or missing some information. We agree with the reviewer that the sentence “"which properties of experience sharing improve multi-level innovation"” was not clear. By “properties of experience sharing” we actually referred to the behavioral and mnemonic metrics we introduced to monitor how the behavior and memories of the agents change with social network structure and task. But as we had not mentioned these metrics before this sentence in the abstract, readers could not understand this. We have therefore rewritten the abstract to properly discuss these metrics, have removed this sentence “"which properties of experience sharing improve multi-level innovation"” and have added the sentence “ to test the hypothesis that the social network structure affects the performance of distributed
> RL algorithms” to address Point 7.
>
> This is our revised abstract:
>
> *"The human cultural repertoire relies on innovation: our ability to continuously
> explore how existing elements can be combined to create new ones. Innovation
> is not solitary, it relies on collective accumulation and merging of previous solu-
> tions. Reinforcement learning approaches commonly assume that fully-connected
> topologies are best suited for innovation. However, human laboratory and field
> studies have shown that hierarchical innovation is more robustly achieved by dy-
> namic social network structures. In dynamic settings, humans oscillate between
> innovating individually or in small clusters, and then sharing outcomes with oth-
> ers. To our knowledge, the role of social network structure on innovation has not
> been systematically studied in reinforcement learning. Here we use a multi-level
> problem setting (WordCraft), with three different innovation tasks to test the hy-
> pothesis that the social network structure affects the performance of distributed
> RL algorithms. We systematically design networks of DQNs sharing experiences
> from their replay buffers in varying structures (fully connected, small world, dy-
> namic, ring) and introduce a set of behavioral and mnemonic metrics that extend
> the classical reward-focused evaluation framework of RL to offer more insights.
> Comparing the level of innovation achieved by different experience-sharing social
> network structures across different tasks shows that, first, consistent with human
> findings, experience sharing within a dynamic structure achieves the highest level
> of innovation across tasks. Second, experience sharing is not as helpful when
> there is a single clear path to innovation. Third, the metrics we propose, can help
> understand the success of different social network structures on different tasks,
> with the diversity of shared experience on an individual and group level lending
> the most insights.
> "*

---

> ### Author Response · Authors · 2022-11-17
> **Reply to Point 4 "Paper is ... phenomena occur"**
>
>  As we explained to our previous reply [Reply to point 1 and 2](https://openreview.net/forum?id=BO5_Lm7iD_&noteId=7B4Uo1CTez),  we have revised the manuscript to clearly state our contributions. In particular, we have introduced a new paragraph in “Section 5. Discussion and Future work” with concrete contributions and proposals for future work in distributed RL and cognitive science. We have also removed the broad implications paragraph. We are particularly happy that the reviewer foresees many opportunities to investigate further and hope they find our suggestions for future work useful.

---

> ### Author Response · Authors · 2022-11-17
> **Reply to Point 5: "I am consused... algorithm?"**
>
> In our original submission we referred to SAPIENS as a learning algorithm. In the revised draft we refer to it as a learning framework. We made this change based on this comment of Reviewer DXUd and a comment we received by reviewer Gs4s related to positioning our work in the RL literature. (See our reply [“Reply to Point 5 “[major]”:”](https://openreview.net/forum?id=BO5_Lm7iD_&noteId=_CwofOIL-rR)).
>
> SAPIENS is a learning framework within the distributed Rl framework with the special properties that there is a social network structure and that all agents are both learners and actors. Our version with DQN agents exchanging experience tuples is an algorithmic instantiation within SAPIENS, but we can imagine using other off-policy algorithms or sharing other types of information. We have made the following changes to reflect this:
>
> In Section “2.3 Learning framework”
>
> *"Thus, SAPIENS is a distributed RL learning paradigm where all agents are both actors and learners,
> a setting distinct from multi-agent RL (Garnelo et al., 2021; Christianos et al., 2020; Jiang et al.,
> 2020a), where agents co-exist in the same environment and from parallelised RL (Steinkraus et al.,
> 2005), where there need to be multiple agents. It should also be distinguished from distributed RL
> paradigms with a single learner and multiple actors (Horgan et al., 2018; Espeholt et al., 2018; Nair
> et al., 2015; Garnelo et al., 2021), as multiple policies are learned simultaneously."*
>
> In Section “5. Discussion and future work”:
>
> *"Aside this, the effect of social network structure in distributed RL can be
> extended beyond evolutionary strategies (Adjodah et al., 2019) and beyond our current instantiation
> of SAPIENS, by considering other off-policy algorithms than DQNs and other types of information
> sharing. "*

---

> ### Author Response · Authors · 2022-11-17
> **Reply to Point 6: "two metrics ... causal evidence however"**
>
> This is a sentence in the abstract that we have rephrased in the revised abstract (which we quoted in our [reply to point 3](https://openreview.net/forum?id=BO5_Lm7iD_&noteId=JJMTj3GkTpN)). We also have avoided using words like “explain” that evoke a causal understanding as we do not have a causal analysis. Our methodology for analysing our empirical results is performing ANOVA tests for multiple comparisons and Tukey range tests for pairwise comparisons (this is mentioned in the revised first paragraph of “Section 3. Empirical results”). Therefore, we only detect correlation, as previous human and computation studies have done (Mason et al, 2008; Mason et al, 2012; Derex and Boyd, 2016; Cantor et al, 2021). We should also note that related works in distributed RL traditionally do not test neither for causation nor for correlation (Horgan et al, 2018’ Nair et al, 2015; Garnelo et al, 2021;Scmitt et al, 2019), arguably due to the large computational complexity of experiments, so we hope that our study sets an example in this regard.

---

> ### Author Response · Authors · 2022-11-17
> **Reply to the comments on "clarity & quality (mostly)" - part A**
>
> We thank the reviewer for the many observations made in this list. We have made the corresponding changes to most points and below we comment on the points that merit some discussion:
>
> **Point 7: “The phrase "deceptive game" is misleading. The games do not have agency. Better to choose something like difficult, or non-obvious, or similar.“**
>
> Our use of the name “deceptive game” and “deceptive task” comes from the RL literature, which often refers to tasks with local optima as deceptive (see a few examples (Bontrager et al, 2019); (Anderson et al, 2018), (Ecoffet et al, 2019), (Wang, Lehman et al, 2019)).  And while we understand why the reviewer raises the concern that games do not have agency, we want to note that [agency is not part of the definition of the word deceptive](https://dictionary.cambridge.org/dictionary/english/deceptive). Based on the above, we prefer to refer to “deceptive tasks” because it helps us position them  in the literature concerned with local optima (while a task may be “difficult” for other reasons, such as a large exploration space )
>
> **Point 8:"Specifically, methods ring, small-world, dynamic and fully-connected are instantiations of SAPIENS for different social network structures with 10 DQNs. " Confusing sentence. Are the italics important? Are these methods?”**
>
> In the original draft we were using italics for some of the methods, which was confusing. In the revised draft we have removed our use of italics, as most RL papers refer to the different methods they benchmark without any formatting. The methods compared in our work are: ring, small-world, dynamic, fully-connected, single, no-sharing, A2C and Ape-X. The first four are instantiations of the learning framework SAPIENS.

---

> ### Author Response · Authors · 2022-11-17
> **Reply to the comments on "clarity & quality(mostly) - Part B**
>
> **Point 9: "1.“When reporting p values, please report the full details that went into the calculation, including the means (and variances as appropriate), test statistic, degrees of freedom, the name of the test used, etc. “, 2."In contrast, fully-connected, A2C and Ape-X perform significantly worse. " This claim needs to be quantified.” 3. “We denote statistical significance levels with asterisks.)" This is not acceptable. Please state explicitly what the number of stars reflects. Also, given the enormous number of tests is there a preregistered evaluation plan? Have the analyses been corrected for the number of tests? Also, it is not clear that the trailing parenthesis is closing anything” 4. "While it is not surprising that a single agent with epsilon-greedy exploration can efficiently solve this task, it is not clear at first glance why experience sharing harms performance. Notably this phenomenon has been observed in related works (Souza et al., 2019; Schmitt et al., 2019)." Is this intended to explain the surprise? It would be nice to provide some explanation for the observations. 5. “"Perhaps surprisingly, the fully-connected group does not have the highest conformity (Figure 5, Left). We therefore conclude that there is an upper threshold for connectivity, beyond which shared experiences destabilize learning." Why is the conclusion warranted? I do not follow the logic. “ 6. “"Finally, when looking at the group diversity in Figure 6, fully-connected ranks last and the dynamic topology ranks first. This is rather surprising: the" It would be nice to have experiments designed to test the observation”"**
>
> Above we have grouped the reviewers comments on our statistical analysis and empirical results. As both reviewer DXUd and Gs4s found Section “3. Empirical results” lacking in quality, we have done a major revision on it, so we kindly ask reviewer DXUd to reread it in the revised manuscript. In particular we have implemented the following changes:
>
> We added a description of our statistical testing procedure (first paragraph). In it we mention that we are using ANOVA and Tukey range test. We provide additional information about the statistical testing setup in Appendix E (eg on what the asterisks mean), including tables 2,3,4 revised to include both means and standard deviations (these tables were present in our original submission but included only means). We updated our discussion of Figure 4 with explanations of the samples we compare, the performance metrics we use and the p-values for significant comparisons (Section 3.1). We performed additional tests and reported  p-values for all claims of the form “method A outperforms method B …” in Sections 3.1, 3.2 and 3.3. These sections have been restructured to follow the form: “Hypothesis from cognitive science about performance” - “Our empirical observations about performance”- ”Our investigation of behavioural and mnemonic metrics to understand our empirical observations”.
>
> Thus, to explicitly answer the above points:
>
> 1. We provide details about p-values in the first paragraph of Section 3 and appendix E. Tables 2, 3 and 4 contain the means and variances for all metrics computed in our experiments for the single-path, merging-paths and best-of-ten paths task respectively.
> 2. We have quantified this claim in  “3.1 Overall comparison”  with the following sentence: *“In the merging paths task there were significant differences among methods both for group success $S$. ($ p = 0.4e^{−4}$) and convergence speed $T^+ (p = 0.0095)$. The group success of dynamic ($S = 0.65$) is significantly higher compared to Ape-X ($S = 0.05, p = 0.001$), A2C ($S = 0.0, p = 0.00101$), fully-connected ($S = 0.0, p = 0.00101$) and ring ($S = 0.2, p = 0.0105$)*.
> 3. This information is now provided in the first paragraph of Section 3 (*"where we also indicate statistically significant pairwise comparisons with asterisks (more asterisks denote higher significance)."*) and in appendix E (*"We report
> the exact p values of theses tests in the text and, when applicable, illustrate them in figures using
> a set of asterisks whose number indicates the significance level (p <= 0.05: *, p <= 0.01: **,p <= 0.001: ***, p <= 0.0001: **** )"*).
> 4. We have rewritten Section “3.2 Task: single path”: to address this point. We explain that previous works have observed that experience sharing degrades performance in RL, but this is not the explanation. We hypothesize is that the explanation is that experience sharing destabilizes agents, which we can see through the diversity and volatility metrics.

---

> > ### Author Response · Authors · 2022-11-17
> > **Continue**
> >
> > 5. We have removed this sentence from the revised  Section “3.2 Task: single path” and replaced it with the point above
> > 6. This sentence has been replaced by the sentence: *“Another interesting observation in this task, as well as in the single-path task is that, as we reported above, dynamic exhibits significantly higher group diversity than no-sharing and it is the only structure to do so: this indicates that shared experiences can foster group exploration.”* In general we removed the word “surprising” as it was not clear whether we meant “surprising a priori” (which was the case here) or “surprising after seeing our results” ( our revised draft does not have this kind of surprise)

---

> ### Author Response · Authors · 2022-11-18
> **Reply to Point 10: "we hypothesize that ...  how could it be tested? "**
>
> We agree with the reviewer that our phrasing of this observation was not very clear. Before we explain how we rephrased this sentence, we provide some background: prioritized experience sharing has been introduced in previous works ((Souza et al., 2019; Horgan et al., 2018). It  adds a prioritzed replay buffer (Schaul et al., 2016) and agents use these priorities to weight the probability of sharing an experience, thus making experience sharing also prioritized. We included this appendix to study a second selection mechanism (other than random sampling) and designed our study so that we can disentangle the effect of using prioritized experiences just for individual elearning (no-sharing) and for both individual learning and sharing (SAPIENS methods). To our knowledge, previous works have not tested for this.
>
> We have rephrased our discussion of Appendix E.6 as follows:
>
> *“b) observed a drop in performance under prioritized experience sharing (Souza et al., 2019; Horgan et al., 2018), where the DQNs employ prioritized
> replay buffers (Schaul et al., 2016) and experiences with higher priority are shared more often (see
> Appendix E.6). In agreement with previous works (Souza et al., 2019), we observe that performance
> degrades in all methods that share experiences. This does not happen for no-sharing, which indicates that prioritized experiences are detrimental only when they are shared. To address this, agents
> can recompute priorities upon receiving them from other agents to ensure they agree with their own
> experience (Horgan et al., 2018)”*

---

> ### Author Response · Authors · 2022-12-01
> **Does Reviewer DXUd have any comments on the revised manuscript?**
>
> Dear Reviewer DXUd,
>
> Thanks again for the time spent on our manuscript and your valuable feedback. You concluded your review with the sentence "I believe this could be an excellent and pathbreaking paper, given substantial work to focus, edit, revise, and clarify. " As we have revised the paper to address your questions and comments, we hope that your concerns have been addressed.  Did you have the opportunity to read the new manuscript and our replies? Is there something else you would like us to clarify or address? Generally, please let us know if you have any additional comments.

---

### Official Review · Reviewer_Gs4s · 2022-10-28

**Confidence:** 4
**Correctness:** 2
**Technical Novelty And Significance:** 3
**Empirical Novelty And Significance:** 3
**Recommendation:** 3

**Clarity, Quality, Novelty And Reproducibility:**

The quality of the manuscript is mixed: the central proposal, task, and study design are all strong points in favor of research quality, but are undercut by substantial weaknesses in the statistical analysis used to infer the effectiveness of the proposed algorithm. See detailed comments in prior comment for more detail.

Similarly, the clarity and novelty of the current manuscript leave room for improvement. As identified in the prior section, the manuscript does not sharply communicate its relationship to prior psychology and social science research, to the detriment of claims made throughout the paper.

**Strength And Weaknesses:**

I appreciated the chance to review this submission. This work has several core strengths:
 - The central idea—the application of network structure to group-level exploration and exploitation—is solid, and will likely have a positive impact on the field.
 - As the manuscript identifies, the cross-disciplinary connection with psychology and behavioral research provide strong initial support for this approach.
 - The manuscript provides an interesting environment to explore innovation. The configurability of Little Alchemy lends itself well to studying problems with different optima structures.
- The experimental design is clean and can provide the empirical evidence needed to evaluate the proposed algorithm.

However, I see multiple countervailing weaknesses. I’m enthusiastic about this direction, and so will try to provide specific suggestions for improvement and questions to guide revision. The following areas could be improved:
 - [major] The manuscript needs to clarify and sharpen its connection to prior psychology research.
   - The fact that the current draft draws inspiration from cross-disciplinary research insights (and especially ones so relevant for population-based methods in RL) is a strength. A recurring claim throughout the submission is that its “contributions provide a better understanding of results originally obtained in human experiments.” I’m skeptical of this claim. How specifically do the RL findings from the present studies answer unanswered questions in psychology research? The manuscript does not effectively review and synthesize the modern state of knowledge in psychology, sociology, and organizational research on the topic. In reviewing work concerning human innovation, the manuscript samples heavily from recent work (e.g., Migliano & Vinicius, 2022; Momennejad et al., 2019; Momennejad, 2022), to the exclusion of important, foundational psych work on this particular problem, namely Mason, Jones, & Goldstone (2008), Mason and Watts (2011), and related organizational research like Fang, Lee, & Schilling (2010). (The current draft does well by including Lazer and Friedman.)
   - With a proper review, it is hard to agree with the claim that this paper contributes to the psychology literature on human innovation, as the patterns observed in the present experiments are novel for single-agent RL, but already well-established in psychology and organizational research (e.g., “Our results show that topologies with low initial connectivity [...] performs [sic] best here by improving the exploration of different innovation paths”). More argumentative work is needed to establish a connection flowing from the present RL research to benefits for the psychology literature.
 - [major] The analyses and results that justify claims throughout section 3 are in need of substantial improvement. The current analysis does not do justice to the submission’s strong task and experimental design. Generally, the current version of the manuscript underspecifies its statistical analyses, including key details such as the data being used and the test being run.
   - For example, the first paragraph of results states that “the performance of the dynamic structure is significantly better than all other baselines except for no-sharing (p-value 0.22) and small-world (p-value 0.07)”. What definition of performance are we using (especially given the four different metrics described in section 2.4)? At what point during learning are we measuring performance? What test are we using to derive statistical significance? Considering the last question, the first paragraph of section 3 mentions the use of “the Welch test”. I’m guessing this refers to a Welch’s t-test (though am not 100% sure), which compares two different samples. However, the authors are comparing *eight* different samples. I would not advise using t-tests in this situation, especially without correcting for multiple comparisons (I can’t find any details about a correction for multiplicity). Instead, the gold standard (parametric) approach would be to run an ANOVA and subsequently apply a comparative analysis like Tukey’s range test (with a correction for multiple corrections).
   - As another example, section 3.2 includes the claim, “Perhaps surprisingly, the fully-connected group does not have the highest conformity (Figure 5, Left). We therefore conclude that there is an upper threshold for connectivity, beyond which shared experiences destabilize learning”. Looking at Figure 5, I’m fairly skeptical of this inference. The error bands in this figure (and many of the other figures with training curves) overlap heavily, making it difficult to believe such comparative claims. What statistical model can help provide empirical support for this claim? I think if the authors want to retain the the second sentence about the upper threshold for connectivity, “conclude” should be revised to “hypothesize”.
   - I’ve chosen these two claims as examples, but broadly I’d encourage the authors to revisit their results section and design specific inferential tests for each of their hypotheses / claims (with corrections for multiple comparisons, where appropriate).
 - [major] The manuscript overstretches its claims in the conclusion. While I really like the overall approach, a key limitation for making general claims is the focus on one task / environment, the limited number of experiments and parameters, and the use of a single algorithm. I’d push the authors to gather much more evidence before make broad claims (e.g., “Based on our experimental results, we can provide general recommendations on which topology to use for which task class”), particularly seeking to test the generality and boundary conditions of the improvements they observe.
 - [major] The $p_s$ and $L_s$ parameters likely matter a lot, in combination with learning rate and other parameters / elements of algorithmic design. I might have missed it, but I didn't see any experiments testing the effects of changing these parameters, or text discussing their likely effect on performance. Similarly, the experiments take one particular approach to implementing a "dynamic" network. It’s easy to imagine many different approaches here, including adaptive structures that optimize for some of the metrics the manuscript discusses in Section 2.4. I’d encourage the authors to discuss the non-exclusitivity of their dynamic structure at some point in the manuscript.
 - [major] The manuscript repeatedly proposes a connection between its algorithm and “multi-agent” as a concept, including through “multi-agent topologies” and the “A” in SAPIENS (“multi-Agent”). However, I think it’d be much clearer for readers to discuss the proposed algorithm and current experiments as single-agent, since they take place in a single-agent RL task. The algorithm that the manuscript introduces is *technically* multi-agent, in the same way that classic population-based training (PBT) for single-agent tasks is “multi-agent”. Making a multi-agent connection in these situations is at best somewhat confusing, and at worst misleading. (I think it’s notable that PBT is rarely referred to as multi-agent, except when applied to multi-agent tasks; e.g., tasks that involve two or more agents when computing reward.) The potential for confusing or misleading readers emerges from the large body of multi-agent reinforcement learning research that *does* consider the effects of network structure, in contrast to recurring claims in the manuscript that RL studies “have not to date considered the effect of group connectivity”. Group connectivity was central to the AlphaStar league, for example (Vinyals et al., 2019), and is a common topic of research in the AAMAS community (e.g., Adjodah et al., 2020; Du et al., 2021; Garnelo et al., 2021). Consider how the discussion proposes “scaling up [the] study by applying SAPIENS in environments commonly employed by the multi-agent reinforcement learning community to study innovation”. How does this differ from the prior, parenthetical AAMAS references? Minimally, the discussion’s proposal should include those details. Overall, I strongly recommend reframing and revising the manuscript to clarify the single-agent nature of the task and to avoid this source of confusion.
 - [minor] Relatedly, PBT is a popular distributed framework that sharpens a learning agent’s exploitation (or, arguably, directs its exploration). PBT could be modified to instead sharpen exploration if the update rule took a non-fully connected approach, rather than copying policies from anywhere in the population of agents. I think it’d be worth including Jaderberg et al. (2019) in the review of “most solutions [that adopt] a fully-connected” approach.
 - [minor] On the behavioral metrics, I am skeptical about measuring conformity by examining the “percentage of agents [...] that followed the same trajectory”. If I understand the game correctly, trajectories can trivially differ by alternating between different branches, with no effect on the number of moves made, the ultimate score, or (in retrospect) the states visited. How would the conformity measure react to two agents in the merging-paths task that follow these paths: [A_1, B_1, A_2, B_2, C_1] and [A_1, A_2, B_1, B_2, C_1]?
 - [minor] The societal implications section is extremely broad, and does not ground its general claims. For example, the section does not justify or explain how the method for single-agent RL introduced here (or the specific results, above and beyond the knowledge already accumulated in the psychological sciences) will help with human-AI, AI-AI, or human-human cooperation. As discussed above, the innovation game is a single-agent task; if the agents are not knowingly or voluntarily exchanging experiences, are they really “cooperating” on the task? How do the current results, above and beyond prior psychology knowledge, contribute to human-AI collaboration? Similarly, much more argumentative work is needed to explain the algorithm's connection with “climate catastrophes and global pandemics”.

**Summary Of The Paper:**

This manuscript leverages insights from the psychological and behavioral sciences to introduce a new algorithm improving exploration among a population of reinforcement learning agents. The psychological insights concern the effect of network structure on the effectiveness of group-level exploration and exploitation, as a function of task structure. The manuscript describes the algorithm and evaluates its effectiveness on an innovation task, applying several different network structures. The manuscript interprets the results of these experiments as evidence for the effectiveness of the algorithm over several reasonable baseline.

**Summary Of The Review:**

Overall, the core idea is solid and backed by a wealth of evidence from psychology and related fields. However, my enthusiasm for the underlying proposal is tempered by the current version of the content, particularly the background review, statistical analysis, and discussion claims. Generally, I suspect the manuscript could be substantially improved with a thorough re-write and additional time spent on statistical analysis of the current experiments (with little-to-no need for additional experiments, perhaps aside from parameter robustness experiments, if the authors wanted to build more evidence for generality).

---

> ### Author Response · Authors · 2022-11-14
> **Reply to Point 1: "[major] The manuscript needs ...  benefits for the psychology literature " - part A**
>
> We agree with the reviewer that the previous draft did not contain a complete review of the cognitive science literature and our contributions to it. We would like to thank them for their suggested references, but also note that the submitted version contained the reference Mason and Watts (2011), mentioned by the reviewer (correct us if you are not referring to the paper with title ‘Collaborative learning in networks’) and reference (Lazer, 2007) seems to us very similar, in terms of methodology and conclusions to (Fang, Lee, & Schilling (2010)), suggested by the reviewer (but the latter contains additional interesting results so we added it in the updated version). Below we break our reply into how we addressed the two suggestions made by the reviewer in this comment: a) clarifying our connection to prior psychology research and b) explaining how we contribute to the psychology literature on human innovation
>
> a) Clarifying our connection to prior psychology research.
>
> In our previous introduction we cited studies in psychology, cognitive science and ecology to explain the hypotheses that motivated our work, namely that social network structure influences collective innovation and that partially connected structures improve performance. Although we did not aim for a comprehensive review of these fields, we cited a diverse set of works with human lab studies, computational and theoretical models (Derex & Boyd, 2016; Cantor et al., 2021,Mason & Watts, 2012; Mason et al., 2008; Lazer & Friedman, 2007; Fang et al., 2010, Sol e et al., 2013). However, we agree with the reviewer that something was missing to connect our work to the current state of these fields. To address this, we made a summary of open challenges in current psychology research, where we focus on two points: i) the fact that the literature has studied a variety of collective search tasks, but few works have studied innovation tasks, i.e., collective search tasks with a multi-level search space and rewards monotonically increasing with levels (we formulate this definition in the second paragraph of Section 1. Introduction). What is more, the works that study innovation  (Derex & Boyd, 2016; Cantor et al., 2021)  have considered a single type of innovation task that inspired our merging-paths task, while, as we show by introducing the single-path task and best-of-ten paths task, more types are possible and they lead to different conclusions on the role of social network structure. ii) the fact that laboratory studies have collected data about the behaviors but not the memories of humans acting in a group.
>
> In particular, to address this comment, we have added the following paragraph in the introduction:
>
> “ *Despite progress on multiple fronts, many open questions remain before we get a clear understanding of how social network structure shapes innovation. On the cognitive science side, computational and human laboratory studies of collective innovation are few and have studied a single type of innovation task where two innovations are combined to create a new one (Derex & Boyd, 2016; Cantor et al., 2021), while a big part of the literature has studied other types of collective search that do not resemble innovation (Mason & Watts, 2012; Mason et al., 2008; Lazer & Friedman, 2007; Fang et al., 2010). Furthermore, laboratory studies have collected purely behavioural data (Mason et al.,2008; Derex & Boyd, 2016), while studies of collective memory have shown significant influence of social interactions on individual cognition (Coman et al., 2016). This lack of mnemonic data makes it hard to further analyze the effect of social network structure on collective search.On the side of the distributed RL community, studies are hypothesizing that the reason why groups outperform single agents, not just in terms of speed, but also in terms of final performance, is the increased diversity of experiences collected by heterogeneous agents (Nair et al., 2015; Horgan et al., 2018). Nevertheless, studies do not measure this diversity. In this case two
> steps seem natural: introducing appropriate metrics of diversity and increasing it, not only through heterogeneity, but also through the social network topology.* ”
>
> As we also received a comment by reviewer Gs4s about positioning our work in the RL literature, we deemed it useful to include a table for summarizing the works we have reviewed on the subject “How does social network structure affect collective innovation”, which includes works both from cognitive science and RL. Our objective there is to present our literature review in a format that allows one to quickly identify similarities and differences within and across fields. This is Table 1 in Appendix B of the updated manuscript.

---

> > ### Author Response · Authors · 2022-11-14
> > **Replay to Point 1: "[Major] The manuscript needs ... benefits for the psychology literature." - Part B**
> >
> > b) Explain how we contribute to the psychology literature on human innovation
> >
> >  In the introduction of the previous draft, we included a paragraph “Contributions”, which included a point relevant to how our proposal of using DRL agents to study collective innovation can be useful for human studies was: iii) by using the replay buffer as a proxy of the memories of agents, we can directly measure properties such as diversity and alignment of experiences that are challenging to estimate with human studies. However, the contributions we discussed in this paragraph were methodological, while the reviewer is stressing out that we did not discuss how our empirical observations contribute to cognitive science. In particular, the reviewer mentions: “as the patterns observed in the present experiments are novel for single-agent RL, but already well-established in psychology and organizational research”. We respectfully disagree with this statement and have added the following paragraph in Section 5 of the updated manuscript to clarify how the patterns observed in our study can contribute to future human studies :
> >
> > *“We hope that our work will contribute to the fields of cognitive science and DRL in multiple ways. First, our empirical observations of the behavior of different topologies in the single path and best-of-ten-path tasks provide concrete hypotheses for future experiments studying human collective innovation, which has so far been studied only in a task that inspired our merging-paths task (Derex & Boyd, 2016). By continuing the dialogue that has been initiated between human and computation studies (Fang et al., 2010; Lazer & Friedman, 2007; Cantor et al., 2021) to include DRL methods, we believe that cognitive science will benefit from tools that, as we show here, can learn in realistic problem set-ups and can be analyzed not just in terms of their behavior, but also in terms of their memories. Second, we hope that studies in distributed RL will extend their evaluation methodology by analyzing not just rewards, but also behavioral and mnemonic metrics such as diversity, conformity and volatility that, as we show here, correlate with success. Aside this, the effect of social network structure in distributed RL can be extended beyond evolutionary strategies (Adjodah et al., 2019) and beyond our current instantiation of SAPIENS, by considering other off-policy algorithms than DQNs and other types of information sharing. Finally, considering the effectiveness of the dynamic topologies observed in this study, we envision future works that investigate more types of them, as well as meta-learning or online-adaptation versions of SAPIENS where the social network structure is optimized for a desired performance metric.”*

---

> ### Author Response · Authors · 2022-11-14
> **Replay to Point 2: " [major] The analyses and results ... appropriate)."**
>
> We agree with the reviewer that the Results section of the submitted version
>  was lacking in the analysis of the data and have, therefore, made significant changes this section to address the questions raised in this comment and a similar comment made by reviewer GCRF. Specifically for the statistical tests, we were indeed using the Welch t-test, recently used in other works to compare DRL algorithms (Colas et al,  2018), but in different experimental setups. We have replaced it with the ANOVA test and subsequent pairwise comparisons with the Tukey’s range test, as we agree with the reviewer that this is what our study requires due to multiple comparisons. We kindly ask the reviewer to read the results section of the updated draft, where we would like to emphasise the following changes:
>
> * We added a description of our statistical testing procedure (first paragraph)
> * We updated our discussion of Figure 4 with explanations of the samples we compare, the performance metrics we use and the p-values for significant comparisons (Section 3.1)
> * We performed additional tests and reported  p-values for all claims of the form “method A outperforms method B …” in Sections 3.1, 3.2 and 3.3. These sections have been restructured to follow the form: “Hypothesis from cognitive science about performance” - “Our empirical observations about performance”- ”Our investigation of behavioural and mnemonic metrics to understand our empirical observations”.

---

> ### Author Response · Authors · 2022-11-14
> **Reply to Point 3: "[major] The manuscript overstretches ... they observe " and Point 8: "[minor] The societal implications .. "**
>
> We received a similar comment by reviewer GCRF and agree with both of them that the implications stated in the Discussion section were too broad considering the present study. In the updated version, we have a new concluding paragraph regarding our contributions to future cognitive science and RL research, that we discussed in detail in our reply to Comment 1 of reviewer Gs5s. We believe that this paragraph captures the implications of our study and have therefore removed two previous ones on contributions and societal implications.

---

> ### Author Response · Authors · 2022-11-14
> **Reply to Point 4: "[major] The $p_s$ and $L_s$ ... in the manuscript"**
>
> **About parameters $p_s$ and $L_s$**:
>
> We introduced parameters $p_s$ and $L_s$ to present a general form of  SAPIENS, but we indeed did not evaluate robustness to them. We have not tuned these values, we employ $p_s=1$ and $L_s=1$, which are the default values for having social network structures identical to the ones used in previous human and computational studies. We would like to mention that we do have a robustness analysis in appendices, but this is in terms of the learning rate and discount factor on different social network structures (Appendix E.4) and of the hyperparameters of the dynamic structure (Appendix D). We would also like to note that changing $p_s$ and $L_s$ should not affect our qualitative conclusions about differences between structures, as all structures will be changed to the same value to ensure a fair comparison.
>
> However we agree with the reviewer that these parameters affect the performance of a group and their optimal values depend on the task and learning parameters: a partially-connected structure with extensive sharing can lead to the same group buffer properties with a fully-connected structure with less sharing. To address this comment, we are currently running additional experiments with different values of $p_s$ and $L_s$ and plan to add them in a new appendix and mention them on p. 7 alongside results in other appendices.
>
> **About alternative dynamic structures**
>
> The previous draft actually included in Appendix A.3 a robustness analysis of the dynamic structure that we studied in the main paper, which we call dynamic-Boyd, and another type of dynamic structure that oscillates between a phase of full connectivity and a phase of no connectivity, which we call dynamic-periodic. We referred to these experiments in two places in the main text: a) In “Section 2.3 Learning algorithm”: “We provide more information about dynamic topologies in Appendix A.3, where we study how its behavior changes with different values of its hyper-parameters $T_v, p+v$ and present an alternative type of dynamic topology” b) In “Section 3.1”: “In additional experiments we …. c) analyzed how the performance of the dynamic topology varies with its hyper-parameters and derived suggestions for tuning it (see Appendix A.3)”.
>
> In the updated manuscript Appendix A.3 has been renamed to Appendix "D. Analysis of dynamic network topologies". We summarize it below:
>
> In Appendix D, we  illustrate the two dynamic structures in Figure 7, present the results of the parametric analysis in Figures 8 and 9 and, for dynamic-Boyd explain that: a) for the single path task hyperparameters do not affect performance, as we have already observed that all topologies have solved this task b) in the merging-paths task there are two clear effects:  i) (i) “short visits with of high probability lead to bad performance. As such settings lead to a quick mixing of the population, they lead to convergence to the local optimum (ii) long visits with high probability work well. Due to the high visit probability, this setting effectively leads to topology where exactly one agent is always on a long visit. Thus, it ensures that sub-groups stay isolated for at least 1000 episodes, after which inter sub-group sharing needs to  take place to ensure that the sub-groups can progress quickly” c) “In the best-of-ten paths task (right), this structure has a clear optimal hyper-parameterization: short visits with high probability are preferred, which maximizes the mixing of the group and makes early exploration more effective.”
>
> Overall, we agree with the reviewer that future work should consider more types of dynamic topologies. In this work we have focused on one of them in the main paper and studied another one in the appendix, both inspired from human studies, but we can easily imagine that other types will perform better. We agree that dynamic structures that adapt online to improve performance are of particular interest. We have added the following point in Section “4. Related works” of the updated manuscript:
>
> *"“Here dynamic topologies that are adapted to maximize a group’s reward have been shown to
> maximize strategic diversity (Garnelo et al., 2021) and help the agents coordinate on demand (Du
> et al., 2021). In contrast, our dynamic topologies vary periodically independently of the group’s per-
> formance, which is important for avoiding convergence to local optima.”
> "*
>
> We have also added the following sentence in Section  “5. Discussion: and future work” :
>
> *”Finally, considering the effectiveness of the dynamic topologies observed in this study, we envision future works that investigate more types of them, as well as meta-learning or online-adaptation algorithms  where the social network structure is optimized for a desired performance metric.”*

---

> > ### Author Response · Authors · 2022-11-18
> > **Additional experiments on the effect of $p_s$ and $L_s$**
> >
> > We have performed the additional experiments  regarding the effect of the parameters $L_s$ and $p_s$. These are now in Appendix E.8 and are briefly mentioned in Section 3. In particular, we have compared the performance of dynamic and fully-connected topologies in the deceptive coins task we previously described in Appendix E.7 for three values of $L_s$ and three values of $p_s$ (the reason why we chose this task over the Wordcraft ones is that it is less computationally demanding, as it is implemented in JAX and requires less training timesteps).
> >
> > Here is the discussion of these  new results in appendix E.8:
> >
> > *“In Figure 22 we present group success ($S$) averaged across trials for a parametric analysis over $L_s \in (1,6,36)$ and $p_s \in (0.35, 0.7, 1)$. We observe that the dynamic topology finds the optimal solution across conditions except for a small probability of failure for $(L_s=1,p_s=0.35)$ and $(L_s=1,p_s=0.7)$. These values correspond to low amounts of information sharing. In this case, the dynamic structure becomes more similar to a no-sharing structure: the amount of shared information is not enough to help the agents avoid local optima they fall into due to individual exploration. For the fully-connected topology we observe that performance degrades for high amounts of information (($L=36$,$p_s=0.35$), ($L=36$,$p_s=0.7$), ($L=36$,$p_s=1$)). This is in accordance with our expectation that fully-connected topologies lead to convergence to local optima. Interestingly, this structure performs well when $p_s=1$ and $L_s \leq 6$. Thus, sharing more frequently is better than sharing longer batches:  we hypothesize that this is because longer batches have more correlated data, making convergence to local optima more probable.”*
> >
> > And this is our summary of them in Section 3:
> >
> > *“g) tested for the robustness of SAPIENS methods to the amount of sharing (hyper-parameters $L_s$ and
> > $p_s$ introduced in Section 2.3) in Appendix E.8, where we observe sub-optimal performance for low
> > amounts of sharing in dynamic and for large shared batches in fully-connected structures”*

---

> ### Author Response · Authors · 2022-11-14
> **Reply to Point 5: "[major] The manuscript repeatedly  ... source of confusion"**
>
> We agree with the reviewer that our setting is not multi-agent RL and that adopting the right terminology is a technicality important for avoiding confusing readers. But we want to note that our previous draft did not refer to multi-agent RL and that our topologies are multi-agent, as there are multiple RL agents exchanging experiences. To help clarify the learning setting we actually found the reference (Adjodah et al 2019) mentioned by the reviewer very useful. We quote: “Such distributed algorithms rely on an implicit communication network between the processing units being used in the algorithm.These units pass information such as data, parameters, or rewards between each other, often through a central controller. For example, in the popular A3C [ 17] reinforcement learning algorithm, multiple ‘workers’ are spawned with local copies of a global neural network, and they are used to collectively update the global network. These workers can either be viewed as implementing the parallelized form of an algorithm, or they can be seen as a type of multi-agent distributed optimization approach to searching the reward landscape for parameters that maximize performance.” Thus, SAPIENS is a learning framework that belongs to the more general framework of multi-agent distributed optimization exemplified by algorithms like A3C, that is usually referred to as distrbuted RL. To clarify our learning setting we have:
>
> * Added “... in a distributed RL learning paradigm” in the first paragraph
> * Added the following paragraph in Section “3.2 Learning framework”
> *Thus, SAPIENS is a distributed RL learning paradigm where all agents are both actors and learners. This setting differs from multi-agent RL (Garnelo et al., 2021; Christianos et al., 2020; Jiang et al., 2020a), where agents co-exist in the same environment and from parallelised RL (Steinkraus et al., 2005), where there need to be multiple agents. It should also be distinguished from distributed RL paradigms with a single learner and multiple actors(Horgan et al., 2018; Espeholt et al., 2018; Nair et al., 2015; Garnelo et al., 2021), as multiple policies are learned simultaneously.*
> * Clarified in the newly added Table 1 in Appendix B which field each work belongs to

---

> ### Author Response · Authors · 2022-11-14
> **Reply to Point 6: "[minor] Relatedly, PBT is a popular ... approach"**
>
> This is an interesting observation, we agree with the reviewer that studying social network topologies in population-based training may give promising results. In the updated draft we cite the work “Human-level performance in first-person multiplayer games with population-based deep reinforcement learning” in the first paragraph and also added the following sentence in Section “4. Related works”:
>
> *“In population-based training, another learning paradigm where multiple agents are trained to learn a single policy, policies are compared against the whole population, thus only considering a fully-connected social network structure (Jaderberg et al., 2018)”*

---

> ### Author Response · Authors · 2022-11-14
> **Reply to point 7: "[minor] On the behavioral ... C_1]?"**
>
> The reviewer is right, conformity should not vary for trivial variations of the trajectory followed by an agent. Our previous version wrongly described conformity as "the percentage of agents [...] that followed the same trajectory”. We had actually implemented conformity as the percentage of agents that end up with the same element at the of their episode, regardless of the trajectory they followed. Thus, in the example mentioned by the reviewer we consider that the two agents conform to each other.
>
> To address this we added the following sentence on p. 6:
>
> *"conformity $C_t$ is a group-level metric that denotes the percentage of agents in a group that end up with the same element at the end of a given evaluation trial. Thus, agents conform to each other even if they follow alternative trajectories"*

---

> ### Author Response · Authors · 2022-12-01
> **Does Reviewer Gs4s have any comments on the revised manuscript?**
>
> Dear Reviewer Gs4s,
>
> Thanks again for the time spent on our manuscript and your valuable feedback. We have revised the paper to address your questions and comments. We are happy with the new version; as we explained in our other comments we found your proposals helpful for improving the quality of the manuscript. We hope that it has also addressed your concerns and are wondering if you had the opportunity to read our response. Is there something else you would like us to clarify or address? Generally, please let us know if you have any additional comments.

---

### Official Review · Reviewer_3Lpq · 2022-10-29

**Confidence:** 5
**Clarity, Quality, Novelty And Reproducibility:** The work seems original and quality/n…
**Correctness:** 4
**Technical Novelty And Significance:** 4
**Empirical Novelty And Significance:** 4
**Recommendation:** 6

**Strength And Weaknesses:**

SAPIENS experiments show that dynamic topologies of experience sharing are best suited to solve complex innovation tasks

both multi-agent network topology and task structure affect the performance of SAPIENS. Based on our experimental results, we can provide general recommendations on which topology to use for which task class.

- The single-path task is an instance of a class of tasks with no strong local optima (similarly to long-horizon tasks. results show no benefit of experience sharing

- The paper lays out how the various forms of the network interconnect settings considered performed in tasks that are individual (global and local optima same) or group, such as merging-path task. These exhibits strong local optima that requires explotation a certain point in order to discover the global optimum

The results also show that topologies with low initial connectivity (such as no-sharing, small world and dynamic) performs best here by improving the exploration of different innovation paths. The dynamic topology shows up as the highest performance, allowing different groups to reach the merging innovation level in non-optimal paths before sharing their experience during visits to other groups to find the optimal one. Finally, the best-of-ten task is an instance of a class of tasks with a large search space, many local optima and a few global ones. The results show that the dynamic topology performs best, allowing different groups to first explore different paths, then spread the optimal solution to other groups once discovered.

Below are some of the improvements I will like to suggest

1. why 20 trials (referred in section 3) was deemed sufficient
2. in dynamic network is perf. best because you already ran over the combinations many times and chose best interconnect (app a.3 has some details but unclear which one was used)
3. how many steps were in each trial not indicated
4. If first is the case then how does the conclusion follows (fig 4 sec 3.1):
merging paths task the performance of the dynamic structure is significantly better than all other baselines except for no-sharing (p-value 0.22) and small-world (p-value 0.07)…This indicates that, while learners that do not share experiences manage to solve the task with relative success, learners that share experiences under social networks combining large clustering and small shortest path perform best.
5. Also, why the group diversity changes from Fig 5 to Fig 6 for singlepath task
6. Minor: Figure 2 top row the merged path see first element as 5 rewards written in text but shows up as 8 in figure

**Summary Of The Paper:**

The authors present study of the role of multi-agent topology on innovation towards goal of clarifying which social network structures are optimal for which innovation tasks, and which properties of experience sharing improve multi-level innovation. For multi-level hierarchical problem setting (WordCraft), three different innovation tasks were considered. The design networks of DQNs enables sharing experiences from their re- play buffers in varying structures (fully connected, small world, dynamic, ring). The level of innovation achieved by different setting, shows that, first, consistent with human findings, experience sharing within a dynamic structure achieves the highest level of innovation across tasks. Second, experience sharing is not as helpful when there is a single clear path to innovation. For Third, two metrics we propose, conformity and diversity of shared experience, can explain the success of different social network structures on different tasks.

**Summary Of The Review:**

SAPIENS experiments show that dynamic topologies of experience sharing are best suited to solve complex innovation tasks

both multi-agent network topology and task structure affect the performance of SAPIENS. Based on our experimental results, we can provide general recommendations on which topology to use for which task class.

- The single-path task is an instance of a class of tasks with no strong local optima (similarly to long-horizon tasks. results show no benefit of experience sharing

- The paper lays out how the various forms of the network interconnect settings considered performed in tasks that are individual (global and local optima same) or group, such as merging-path task. These exhibits strong local optima that requires explotation a certain point in order to discover the global optimum

The results also show that topologies with low initial connectivity (such as no-sharing, small world and dynamic) performs best here by improving the exploration of different innovation paths. The dynamic topology shows up as the highest performance, allowing different groups to reach the merging innovation level in non-optimal paths before sharing their experience during visits to other groups to find the optimal one. Finally, the best-of-ten task is an instance of a class of tasks with a large search space, many local optima and a few global ones. The results show that the dynamic topology performs best, allowing different groups to first explore different paths, then spread the optimal solution to other groups once discovered.

Below are some of the improvements I will like to suggest

1. why 20 trials (referred in section 3) was deemed sufficient
2. in dynamic network is perf. best because you already ran over the combinations many times and chose best interconnect (app a.3 has some details but unclear which one was used)
3. how many steps were in each trial not indicated
4. If first is the case then how does the conclusion follows (fig 4 sec 3.1):
merging paths task the performance of the dynamic structure is significantly better than all other baselines except for no-sharing (p-value 0.22) and small-world (p-value 0.07)…This indicates that, while learners that do not share experiences manage to solve the task with relative success, learners that share experiences under social networks combining large clustering and small shortest path perform best.
5. Also, why the group diversity changes from Fig 5 to Fig 6 for singlepath task
6. Minor: Figure 2 top row the merged path see first element as 5 rewards written in text but shows up as 8 in figure

---

> ### Author Response · Authors · 2022-11-17
> **Overall reply to reviewer 3Lpq - Part 1**
>
> We thank reviewer  3Lpq for their positive view of our work  and their suggested improvements.
>
> We would like to note that we have revised our manuscript to address the comments raised by all reviewers, where we have indicated the changes highlighted in blue. We encourage the reviewer to read the new version, but this is not necessary to see how their comments were addressed.  Below we reply separately to each point they raised, explaining which changes it led to and quoting the text added in the revised manuscript in italics, if applicable.
>
> **Point 1: “why 20 trials (referred in section 3) was deemed sufficient”**
>
> Based on our tests for statistical significance (see Section “3. Empirical results”), our claims in the abstract, introduction and discussion sessions are backed up by statistically significant results. We should note that, based on comments from other reviewers, we have updated our statistical testing procedure (see first paragraph in Section 3 and Appendix E) and now report p-values in the text whenever we make comparisons.  We also provide the data used for these comparisons (means and standard deviations) in tables 2,3 and 4 of Appendix E.  For the majority of comparisons that do not show statistical significance, we expect that increasing the number of trials will not change this result, either because the compared methods are indeed very close or the large variances will make the required number of trials prohibitively large. We also want to note that 20 trials is a large number for a deep RL study, where most works report at most 10 trials (For example see “How many random seeds? ”, Colas et al, 2018).
>
> **Point 2: “in dynamic network is perf. best because you already ran over the combinations many times and chose best interconnect (app a.3 has some details but unclear which one was used)”**
>
> We thank the reviewer for asking for this clarification as it helps us comment on the robustness of the dynamic structure.
>
> In our original submission we reported the parameters of the dynamic topology in (what is now) Appendix E.1:
>
> "*In the main paper we employ the dynamic-Boyd topology with $T_v=10$ and $p_v=0.001$ across tasks.*"
>
> The same parameters are used for all three tasks and they have been tuned in the merging-paths task (for more information about our hyperparameter analysis we point the reviewer to what is now Appendix D). Such tuning has ensured that the dynamic topology does not end up acting similarly to a fully-connected one (due to too many and long visits) or to the no-sharing method (due to not enough visiting). Thus, in the best-of-ten paths and the single path tasks the performance of the dynamic without being explicitly tuned.
>
> **Point 3: “how many steps were in each trial not indicated”**
>
> Thank you for asking for this clarification. We have added this information at the beginning of Appendix E:
>
> “*To ensure that all methods have the same number of samples, we assume that, for trials where a method did not find the optimal solution, and, hence, $ T^{+}$ is undefined, $ T^{+}$ is equal to the total number of timesteps the method was trained for, $T_{\text{train}}$. For each task, all methods have been trained for an equal duration of time: $T_{\text{train}}=1 e^6$ for the single path ,  $T_{\text{train}}=7 e^6$  for the merging paths task and $T_{\text{train}}=2 e^7$ for the best-of-ten paths task.*”
>
> **Point 4: “If first is the case then how does the conclusion follows (fig 4 sec 3.1): merging paths task the performance of the dynamic structure is significantly better than all other baselines except for no-sharing (p-value 0.22) and small-world (p-value 0.07)…This indicates that, while learners that do not share experiences manage to solve the task with relative success, learners that share experiences under social networks combining large clustering and small shortest path perform best.”**
>
> The reviewer is correct, this sentence was contradictory. We have removed it from the revised manuscript, where we have in overall improved the discussion of results. In particular, this sentence has been replaced (Section 3.1) by:
>
> *“In the merging paths task there were significant differences among methods both for group success $S$ ( $p=0.4 e^{-4}$) and convergence speed $T^+$ ($p=0.0095$). The group success of dynamic ($S=0.65$) is significantly higher compared to Ape-X ($S=0.05,p=0.001$),  A2C ($S=0.0,p=0.00101$),  fully-connected ($S=0.0,p=0.00101$)  and ring ($S=0.2,p=0.0105$). The single, no-sharing and small-world structures performed comparably well, but did not show statistically significant differences with other methods.”*

---

> ### Author Response · Authors · 2022-11-17
> **Overall reply to reviewer 3Lpq - Part 2**
>
>
> **Point 5: “Also, why the group diversity changes from Fig 5 to Fig 6 for singlepath task”**
>
> Figure 5 contained the individual diversity (diversity of an agent, averaged within the group) while Figure 6 contained the group diversity (we combine all replay buffers in a single group buffer and compute its diversity). (In the updated manuscript Figure 5 has become the left part of figure 5 and Figure 6 has become the right of Figure 5) These two metrics differ a lot: for example in the single-path task, we see that fully-connected has the highest individual diversity but the lowest group diversity. As we say in this revised text (Section 3.2) this is because: *“sharing experiences with others diversifies an individual's experiences but also homogenizes the group”*
>
> **Point 6: Minor: Figure 2 top row the merged path see first element as 5 rewards written in text but shows up as 8 in figure**
>
> We are not completely sure but we believe that the reviewer is referring to the text *“For example, the innovation level of $A_3$ in the single path task is $3$, while the innovation level of element $C_1$ in the merging paths task is 5 ($1$ on C $+$ 2 on A $+$ 2 on B).”*. This text is in agreement with Figure 2: the innovation level appears on the left of a combination (5) and the reward on the right (8, in red bold, as explained in the caption of the figure).

---

### Author Response · Authors · 2022-11-18
**Summary of the reviewer's main concerns and how we addressed them**

We thank all reviewers for their constructive feedback. We have replied separately to all the points they raised and would like to provide a summary of all the comments we received and how we addressed them. We hope that this summary will be helpful for both reviewers and the AC.

We received many encouraging comments from the reviewers, in particular about the:

*  originality and importance of the idea (3Lpq, DXUd, GCRF, Gs4s)
* quality of the experimental design (3Lpq, Gs4s, DXUd, GCRF)
* grounding of hypotheses in cognitive science (Gs4s, GCRF, DXUd)
* future impact of contributions (GCRF, Gs4s)

We also received major and minor suggestions by the reviewers. Of these we would like to emphasise that we were asked to:

1) Improve our literature review of prior psychology research to clarify and sharpen our connection to it and explain how this work contributed to it (Gs4s)
2) Improve our statistical analysis and discussion of results (Gs4s, DXUd, GCRF)
3) Empirically study how $L_s$ (length of shared batch) and $p_s$ (probability of sharing) affect the performance of SAPIENS (Gs4s)
4) Avoid overstretching our contributions (Gs4s, DXUd)
5) More clearly position SAPIENS in the RL literature (Gs4s and DXUd)

We have extensively revised our manuscript based on the above and other comments, indicating changed text in blue. Below we summarise how our changes addressed the above main weaknesses:

1) We have added a paragraph on the limitations of current research in both cognitive science and RL on how social network structure shapes innovation in Section “1. Introduction” and a table summarizing our literature review (Table 1  in Appendix B). We have also explained how our empirical study contributes to future studies in psychology with a new sentence in the paragraph “Contributions” in Section “1. Introduction” and a new paragraph at the end of Section “5.Conclusions”.  Specifically, the contributions to prior psychology research that we mention in the revised manuscript are:

  (i) (methodological): By using deep RL as a computational model for studying hypotheses from human studies we a) enable learning in complex and large search spaces with tasks similar to the ones used in human studies b) adopt a learning mechanism that more closer follows the methodology used in human studies (see Derex and Boyd, 2016) compared to agent-based models of the past (eg see Lazer and Friedman, 2007; Cantor et al, 2021): agents exchange experiences in the form of observations that are stored to memory and are processed by the policy  c) enable measuring mnemonic metrics (this is  possible due to the replay buffer, while previous agent-based models did not have a memory mechanism) (ii) (empirical) Our empirical study has a) confirmed a hypothesis from human experiments with a computational model (namely that a dynamic structure solves the merging-paths task more robustly compared to fully-connected structures b) studied two more tasks and more structures, analyzing mnemonic and behavioral metrics, and derived conclusions that can be used as hypotheses in future human studies.

For our detailed reply to this point, please see our reply to reviewer Gs4s [“Reply to Point 1: …”](https://openreview.net/forum?id=BO5_Lm7iD_&noteId=GySX_9CDQCK)

2. We have done major changes to Section “3. Results”. First, we have replaced the Student t-tests with ANOVA for multiple comparisons and the Tukey range test for pairwise comparisons to account for multiple comparisons. (We explain this in the first paragraph of Section “3. Results” and in Appendix E). Second, we report the means and variances of all metrics in tables 2,3 and 4 of Appendix E for the three tasks. Finally, we have rewritten our discussion of results to improve its quality by reporting statistically significant results and properly explaining how we used our evaluation metrics to derive our conclusions.

3. This was the only comment that required additional experiments, which are now in  Appendix “E.8 Robustness to amount of sharing ($p_s$ and $L_s$)” and are briefly discussed in Section “3. Empirical results”. You can see our [reply to reviewer Gs4s](https://openreview.net/forum?id=BO5_Lm7iD_&noteId=znEiDVhVmt) for more information.

4. We have removed the two paragraphs in Section “5. Discussion and future work” that were mentioning generic contributions to human-human, human-AI, AI-AI cooperation. We have discussed specific limitations and contributions of our experimental study, both from the perspectives of cognitive science and RL, in the last two paragraphs of the manuscript.

5. We have clarified that SAPIENS is a learning framework within the distributed RL framework (new paragraph in Section “2.3 Learning framework”). We have also improved our discussion of related works in RL (see new section “4. Related work” and Table 1 in Appendix B indicating which sub-fields of RL the different works belong to)

---

### Decision · Program_Chairs · 2023-01-20

**Decision:**

Reject

**Justification For Why Not Higher Score:**

While the paper pitches itself as an interdisciplinary work between ML and CogSci, I feel that it does not live up to the standards of either field. On the ML side, the experiments are somewhat toy and it's not clear whether the approach would scale to more challenging RL environments. Despite the author's revisions, the statistics also still feel imprecise which undermines the potential impact. On the CogSci side, the impacts are more theoretical and not demonstrated in practice. Overall, the paper is unfocused and it is not clear what the central claims are. This therefore limits its potential interest and impact at ICLR.

**Justification For Why Not Lower Score:**

N/A

**Metareview: Summary, Strengths And Weaknesses:**

Inspired by the cognitive science literature, this paper proposes a new distributed RL setup in which there are multiple actor/learner processes arranged in a graph topology such that adjacent processes in the graph share experience tuples with each other. The paper explores the performance of multiple different topologies (e.g. fully-connected, ring, small-world, dynamic) in three different tasks (single path, merging paths, and best-of-ten). While in the single path task (where there is a single, optimal path to the goal) all distributed topologies perform comparably well, in the other tasks the non-fully-connected topologies---and in particular the dynamic topology---perform better. The paper performs a number of analyses to better understand why this is the case, and argues that different distributed topologies could be useful for deep RL in general, as well as being useful as a modeling tool in cognitive science.

The reviewers agreed that the core idea of the paper---changing the way that distributed RL agents share experience via different network topologies---was compelling and important. However, the reviewers also agreed that the paper suffered from a lack of focus. During the discussion, the reviewers acknowledged the revisions made by the authors and felt like the paper was improved to some degree (especially in terms of toning down some of the claims, and the statistical analysis), but that a deeper issue remained regarding the focus of the paper. In particular, the reviewers felt that the paper wanted to make a claim about significance both from ML and CogSci standpoints, yet failed to do so convincingly along either dimension (especially on the CogSci side, though I also felt it could be more convincing from the ML side if experiments were performed on more canonical and challenging domains, e.g. Atari). One suggestion that came out of the discussion was to actually split the paper into two: one focused for a ML audience, and one focused for a CogSci audience, which would help sharpen the focus and claims. Regardless, it was agreed that the paper needed a more holistic revision going beyond the additions/modifications done during the rebuttal before it would be ready for publication at ICLR.

I recommend rejection at this stage, and encourage the authors to undertake a more extensive revision of the paper. I think with an improved manuscript that is more precise about the results and impact, this could be a really fantastic piece of work, so I do hope the authors will continue working on it!

**Summary Of Ac-Reviewer Meeting:**

I held a meeting with 3/4 reviewers (one did not show up). One of the reviewers was in favor of accepting the paper, while the other two were in favor of rejecting the paper. All reviewers said that they really liked the core idea/question in the paper, and they felt that the authors did a good job walking back some of their claims in the revision. However, the reviewers who were in favor of rejecting the paper made a strong case that they felt the paper had some fatal flaws in terms of focus and clarity, and that it needed a more holistic revision. By the end of the discussion the reviewer who was in favor of accepting agreed that the paper felt unfocused and could still benefit from revision. In all, the reviewers appreciated the revision by the authors but felt it did not go far enough and that a more substantial revision would be required to address the deeper issues about what the main claims are and how they are evaluated.